# Multistability and dynamic transitions of intracellular Min protein patterns

Fabai Wu[1,†], Jacob Halatek[2,†], Matthias Reiter[2], Enzo Kingma[1], Erwin Frey[2,*] & Cees Dekker[1,**]

## Abstract

Cells owe their internal organization to self-organized protein patterns, which originate and adapt to growth and external stimuli via a process that is as complex as it is little understood. Here, we study the emergence, stability, and state transitions of multistable Min protein oscillation patterns in live *Escherichia coli* bacteria during growth up to defined large dimensions. *De novo* formation of patterns from homogenous starting conditions is observed and studied both experimentally and in simulations. A new theoretical approach is developed for probing pattern stability under perturbations. Quantitative experiments and simulations show that, once established, Min oscillations tolerate a large degree of intracellular heterogeneity, allowing distinctly different patterns to persist in different cells with the same geometry. Min patterns maintain their axes for hours in experiments, despite imperfections, expansion, and changes in cell shape during continuous cell growth. Transitions between multistable Min patterns are found to be rare events induced by strong intracellular perturbations. The instances of multistability studied here are the combined outcome of boundary growth and strongly nonlinear kinetics, which are characteristic of the reaction–diffusion patterns that pervade biology at many scales.

**Keywords** reaction-diffusion patterns; Min protein oscillations; cell shape; cell growth; Turing instability

**Subject Categories** Quantitative Biology & Dynamical Systems; Development & Differentiation

**Mol Syst Biol. (2016) 12: 873**

## Introduction

Many cells have characteristic forms. To guide proper assembly of their subcellular structures, cells employ machineries that garner and transmit information of cell shape (Kholodenko & Kolch, 2008; Shapiro *et al*, 2009; Moseley & Nurse, 2010; Minc & Piel, 2012). But cells are not static objects: They grow, divide, and react to stimuli, and these processes are often accompanied by a change of cell shape. Hence, the means by which a cell gathers spatial information need to be adaptive. One versatile mechanism that is capable of such spatial adaptation is self-organized pattern formation (Cross & Hohenberg, 1993; Epstein & Pojman, 1998; Murray, 2003).

Spontaneous emergence of spatial structures from initially homogeneous conditions is a major paradigm in biology, and Alan Turing's reaction–diffusion theory was the first to show how local chemical interactions could be coupled through diffusion to yield sustained, non-uniform patterns (Turing, 1952). In this way, the symmetry of the starting system can be broken. Reaction–diffusion mechanisms have been shown to account for the generation of many biological patterns (Kondo & Miura, 2010). However, how patterns change in response to noise and perturbations, be they chemical or geometrical, is poorly understood. Resolution of such issues is critical for an understanding of the role of reaction–diffusion systems in the context of the spatial confines and physiology of a cell (or an organism). To include the effects of geometry, the mathematical framework for reaction–diffusion theory has been extended to circular (Levine & Rappel, 2005), spherical (Klünder *et al*, 2013), and elliptical geometries (Halatek & Frey, 2012). However, focusing on pattern formation from homogeneity is not enough, as was noted by Turing himself at the end of his seminal article in 1952 (Turing, 1952): "Most of an organism, most of the time, is developing from one pattern into another, rather than from homogeneity into a pattern".

Min proteins form dynamic spatial patterns that regulate the placement of division sites in prokaryotic cells and eukaryotic plastids (de Boer *et al*, 1989; Hu & Lutkenhaus, 1999; Raskin & de Boer, 1999; Colletti *et al*, 2000; Maple *et al*, 2002; Ramirez-Arcos *et al*, 2002; Szeto *et al*, 2002; Leisch *et al*, 2012; Leger *et al*, 2015; Makroczyová *et al*, 2016). In rod-shaped *Escherichia coli* cells, MinD and MinE form a reaction–diffusion network that drives pole-to-pole oscillations in their local concentrations (Hu & Lutkenhaus, 1999; Raskin & de Boer, 1999; Huang *et al*, 2003). Membrane-bound MinD binds MinC, which inhibits FtsZ polymerization (Dajkovic *et al*, 2008). The dynamic Min oscillation patterns thus result in maximal inhibition of FtsZ accumulation at the cell poles and minimal inhibition at the cell center, which, together with a nucleoid occlusion mechanism, restricts formation of the division apparatus to midcell (Adams & Errington, 2009). Because it exhibits a multitude of complex phenomena, which can be explored by experimental and theoretical means, the Min

1 Department of Bionanoscience, Kavli Institute of Nanoscience, Delft University of Technology, Delft, The Netherlands
2 Arnold-Sommerfeld-Center for Theoretical Physics and Center for NanoScience, Ludwig-Maximilians-Universität München, München, Germany
*Corresponding author. Tel: +49 8921804537; E-mail: frey@lmu.de
**Corresponding author. Tel: +31 152786094; E-mail: c.dekker@tudelft.nl
†These authors contributed equally to this work

oscillator provides an informative reference system for the quantitative study of geometry-responsive pattern formation.

The dynamic Min oscillations have been explained by reaction–diffusion models based on a minimal set of interactions between MinD, MinE, ATP, and the cell membrane (Howard *et al*, 2001; Meinhardt & de Boer, 2001; Kruse, 2002; Huang *et al*, 2003; Fange & Elf, 2006; Touhami *et al*, 2006; Loose *et al*, 2008; Halatek & Frey, 2012). MinD, in its ATP-bound form, cooperatively binds to the cytoplasmic membrane (Hu *et al*, 2002; Mileykovskaya *et al*, 2003). MinE interacts with membrane-bound MinD, triggering the hydrolysis of its bound ATP and releasing MinD from the membrane (Hu *et al*, 2002; Shih *et al*, 2002; Hsieh *et al*, 2010; Loose *et al*, 2011; Park *et al*, 2011). MinD then undergoes a nucleotide exchange cycle in the cytosol, which was initially incorporated into the modeling framework by Huang *et al* (Huang *et al*, 2003). Further theoretical analysis of the minimal reaction scheme suggested that the interplay between the rate of cytosolic nucleotide exchange and strong preference for membrane recruitment of MinD relative to MinE facilitates transitions from pole-to-pole oscillations in cells of normal size to multinode oscillations (striped mode) in filamentous cells (Halatek & Frey, 2012). Such transitions occur if proteins that have detached from one polar zone have a greater tendency to re-attach to the membrane in the other half of the cell rather than to the old polar zone—a process which has been termed *canalized transfer*. This leads to synchronized growth and depletion of MinD from spatially separated polar zones, enabling the simultaneous maintenance of multiple polar zones. Numerical simulations of a reaction–diffusion model based on this canalized transfer of Min proteins successfully explain a plethora of experimentally observed Min oscillations in various geometries (Halatek & Frey, 2012).

Essential for the robust function of Min proteins in ensuring symmetric cell division is their ability to respond to, and thus encode, information relating to cell shape. Upon cell-shape manipulation, Min proteins have been found to exhibit a range of phenotypes under different boundary conditions (Corbin *et al*, 2002; Touhami *et al*, 2006; Varma *et al*, 2008; Männik *et al*, 2012; Wu *et al*, 2015b). Recent development of a cell-sculpting technique allows accurate control of cell shape over a size range from $2 \times 1 \times 1\ \mu m^3$ to $11 \times 6 \times 1\ \mu m^3$, in which Min proteins show diverse oscillation patterns, including longitudinal, diagonal, rotational, striped, and even transverse modes (Wu *et al*, 2015b). These patterns were found to autonomously sense the symmetry and size of shaped cells. The longitudinal pole-to-pole mode was most stable in cells with widths of $< 3\ \mu m$, and lengths of 3–6 μm. In cells of this size range, Min proteins form concentration gradients that scale with cell length, leading to central minima and polar maxima of the average Min concentration. Increasing cell length to 7 μm and above led to the emergence of striped oscillations. In cells wider than 3.5 μm, Min oscillations can align with the short axis of the lateral rectangular shape, yielding a transverse mode (Wu *et al*, 2015b). The existence of various oscillation modes has also been reconstituted *in vitro* with MinD, MinE, ATP, and lipid bilayers confined to microchambers (Zieske & Schwille, 2014). Numerical simulations based on an established reaction–diffusion model (Halatek & Frey, 2012) successfully recaptured the various oscillation modes in the experimentally sampled cell dimensions (Wu *et al*, 2015b). This further emphasizes the role of the two above-mentioned factors generic to reaction–diffusion processes in cells: cytosolic nucleotide exchange and membrane recruitment (Huang *et al*, 2003; Halatek &

Frey, 2012). These data provided the first evidence that sensing of geometry is enabled by establishing an adaptive length scale through self-organized pattern formation.

Given that Min proteins in all cells initially adopt the same regime of pole-to-pole oscillations, it is as yet unclear how diverse oscillation modes emerge during cell growth to large dimensions, and whether transitions occur between these patterns. Furthermore, more than one mode of oscillation was often observed in different cells with the same shape, presenting an intriguing example of the multistability of different complex patterns (Wu *et al*, 2015b). These unexplained phenomena provide us with the rare opportunity to quantitatively explore the basic principles of the dynamics of pattern formation in the context of geometric perturbations and cellular heterogeneities.

In this study, we combine experiments and theory to systematically examine the emergence and dynamic switching of the distinct oscillatory Min protein patterns (longitudinal, transverse, and striped oscillations, cf. Fig 1A) observed in *E. coli* bacteria that are physically constrained to adopt defined cell shapes. Our primary aim was to investigate the origin of multistability (coexistence of stable patterns), and to further understand its relevance in the context of cell growth (i.e. changing cell shape). Furthermore, we hoped to identify the kinetic regimes and mechanisms that promote transitions between patterns and to probe their robustness against spatial variations in kinetic parameters. One striking discovery is the high degree of robustness of individual modes of oscillation even in the face of significant changes in geometry.

To present our results, we first show experimentally that different patterns can emerge out of near-homogeneous initial states in living cells with different dimensions, thus providing further support for an underlying Turing instability. We then use computational approaches to capture the dependence of pattern selection on geometry. Using stability analysis, we establish kinetic and geometric parameter regimes that allow both longitudinal and transverse patterns to coexist. Furthermore, we evaluate the emergence and stability of these patterns in computer simulations and compare the results with experimental data. Remarkably, we find that the experimentally observed multistability is reproduced by the theoretical model in its original parameter regime characterized by canalized transfer. In experiments, we trace pattern development during the cell-shape changes that accompany cell growth, and we quantitatively assess the persistence and transition of patterns in relation to cell shape. These analyses reveal that Min patterns are remarkably robust against shape imperfections, size expansion, and even changes in cell axes induced by cell growth. Transitions between multistable patterns occur (albeit infrequently), driving the system from one stable oscillatory pattern to another. Altogether, this study provides a comprehensive framework for understanding pattern formation in the context of spatial perturbations induced by intracellular fluctuations and cellular growth.

# Results

## Symmetry breaking of Min patterns from homogeneity in live *E. coli* cells

One of the most striking examples of the accessibility of multiple stable states observed in shaped *E. coli* cells is the emergence of

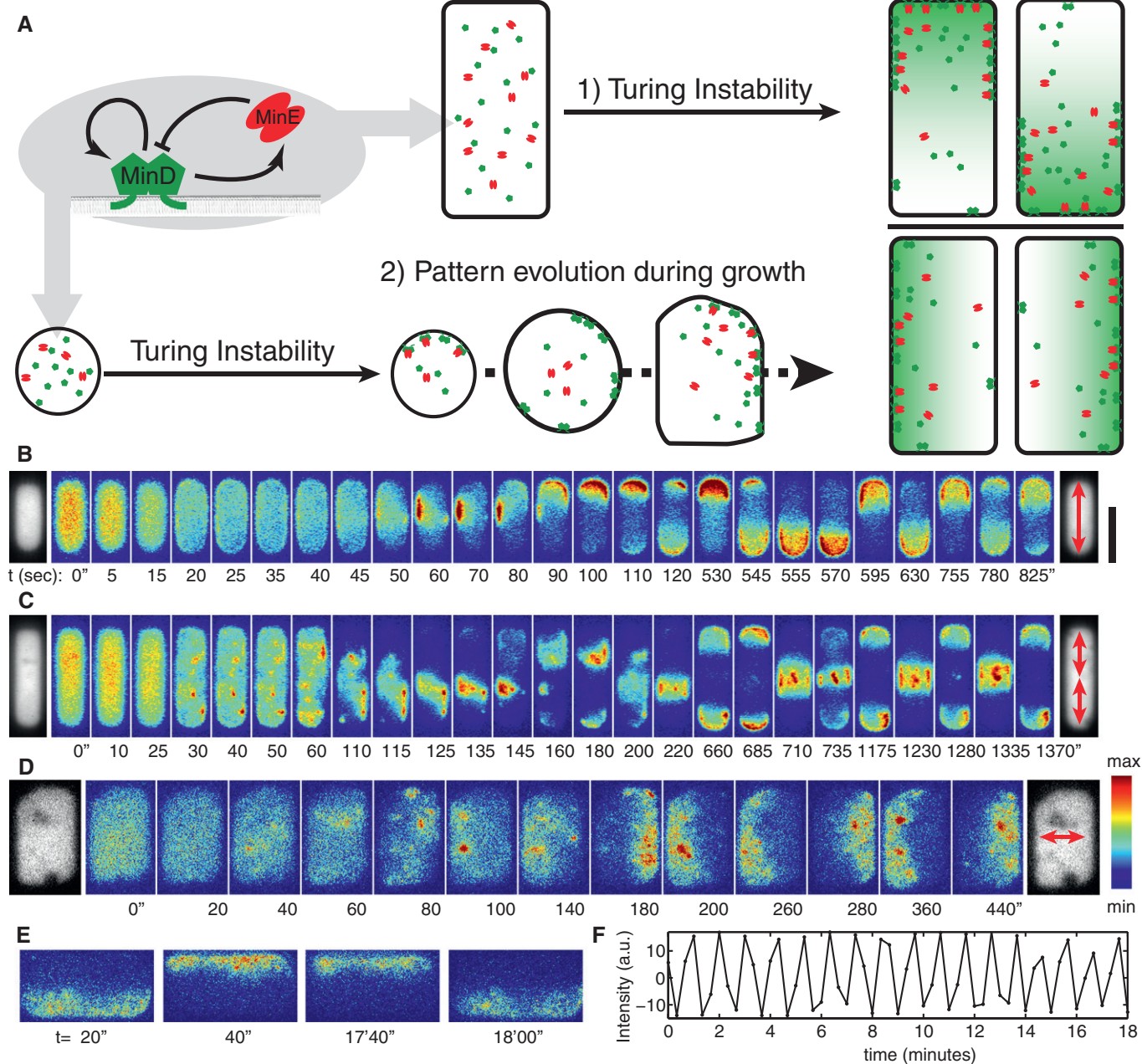

**Figure 1. Symmetry breaking of Min protein patterns *in vivo*.**

A   Schematic showing Min protein patterns in a defined geometry originating from 1) a dynamic instability arising from an equilibrium state or 2) dynamic transitions from a pre-existing pattern associated with cell growth. Green and red particles represent MinD and MinE proteins, respectively. The green gradient depicts the MinD concentration gradient.

B–D   Examples of Min protein patterns emerging from nearly homogeneous initial conditions in *E. coli* cells of different sizes. Lateral dimensions (in μm) from top to bottom: 2 × 6.5, 2 × 8.8, and 5.2 × 8.8, respectively. The gray-scale images show cytosolic near-infrared fluorescence emitted by the protein eqFP670 at the first (left) and last (right) time points. The color montages show the sfGFP-MinD intensity (indicated by the color scale at the bottom right) over time. The scale bar in panel (B) corresponds to 5 μm. Red arrows show the oscillation mode at the respective time point.

E   Two early and two late frames depicting sfGFP-MinD patterns in a cell exhibiting stable transverse oscillations. The images share the scale bar in (B).

F   Difference in sfGFP-MinD intensity between the top half and bottom half of the cell plotted against time.

different—transverse and longitudinal—Min oscillation modes in rectangular cells with identical dimensions (Wu *et al*, 2015b). The existence of a transverse mode has also been noted in reconstituted *in vitro* systems (Zieske & Schwille, 2014). In live cells, this phenomenon is most prominent in cells with widths of about 5 μm and lengths of between 7 and 11 μm (Wu *et al*, 2015b). To probe the emergence and stability of these different stable states, we began this study by monitoring the temporal evolution of Min protein

patterns in deformable cells growing in rectangular microchambers. Improving upon our previous shaping and imaging method (see Materials and Methods), we recorded cytosolic eqFP670 (a near-infrared fluorescent protein) and sfGFP-MinD fluorescence signals over the entire course of cell growth (~6–8 h). Owing to the superior brightness and photostability of these two fluorescent probes (Wu *et al*, 2015a), we were able to image the cells at 2-min intervals without affecting cell growth. Given that an oscillation cycle (or period) takes $68 \pm 13$ s (mean $\pm$ SD) at our experimental temperature (26°C), shorter intervals were subsequently used to capture the detailed dynamics within one oscillation cycle.

We first grew cells with the above-mentioned lateral dimensions $(7–11 \times 5 \times 1 \ \mu m^3)$ in microchambers of the appropriate form. Of the 126 cells examined, almost all ($n = 121$) showed clear MinD polar zones in all times prior to cell death or growth beyond the confines of the chambers, demonstrating the striking persistence of the oscillation cycles. In some cells, transition states between different patterns were also captured, which are described below (see Sections Persistent directionality traps Min oscillations in a stable state during cell growth and Experimental observations of pattern transitions between multistable states). Interestingly, imaging of the remaining five cells captured 1–2 frames in which the sfGFP-MinD fluorescence was distributed *homogeneously* (Fig EV1 and Video EV1). Such a homogeneous state phenomenologically resembles the initial conditions chosen in the majority of chemical and theoretical studies on pattern formation. However, in the present case, Min proteins re-established oscillations *exclusively* in the transverse mode, irrespective of their preceding oscillation mode (Fig EV1). Why the system should "revert" to such a homogeneous state in the first place is unknown, although the rapid recovery of patterns leads us to speculate that it most probably results from a transient effect, such as a change in membrane potential or a rearrangement of chromosomes, rather than from a drastic depletion of ATP. Nonetheless, such an intermittent state provides a unique opportunity to study the emergence of patterns from a spatially uniform background.

We therefore explored symmetry breaking by Min proteins over a larger range of cell sizes and found that different cell dimensions gave rise to different patterns from an intermittent homogeneous state. Because homogeneous distributions of MinD are observed at low frequency, we manually searched for cells in such a state. Once targeted, such cells were subsequently imaged at short time intervals of between 5 and 20 s until an oscillation pattern stabilized. As shown in Fig 1B–D, the uniform distribution of sfGFP-MinD seen in cells of different sizes and shapes became inhomogeneous, and always re-established stable oscillations within a few minutes. In the $6.5 \times 2 \times 1 \ \mu m^3$ cell shown in Fig 1B, the homogeneous sfGFP-MinD signal first became concentrated at the periphery of the cell, indicating a transition from the cytosolic state to the membrane-bound form. At $t = 20$ s, a minor degree of asymmetry was observed. Within the next 30 s, a clear sfGFP-MinD binding zone developed on the left-hand side of the top cell half. This zone persisted for 40 s, until a new binding zone was established at the top cell pole, which then recruited the majority of the sfGFP-MinD molecules. This pattern rapidly evolved into longitudinal pole-to-pole oscillations which lasted for the rest of the time course of our time-lapse imaging (10 min). In an $8.8 \times 2 \times 1 \ \mu m^3$ cell (Fig 1C), the initial membrane binding of sfGFP-MinD was accompanied by

the formation of several local patches of enhanced density (see, e.g. $t = 30$ s), which went on to form one large patch that was asymmetrically positioned in relation to the cell axes ($t = 110$ s). This MinD binding zone further evolved into a few cycles of asymmetric oscillations before converging into striped oscillations, with sfGFP-MinD oscillating between two polar caps and a central striped. In the $8.8 \times 5.2 \times 1 \ \mu m^3$ cell (Fig 1D), persistent transverse oscillations emerged within ~2.5 min after clusters of sfGFP-MinD had begun to emerge as randomly localized, membrane-bound patches from the preceding homogeneous state.

To further examine the stability of the transverse mode, we tracked transverse oscillations in 5-μm-wide cells with a time resolution of 20 s. We found that these indeed persisted, with a very robust oscillation frequency, for at least 17 cycles (i.e. the maximum duration of our experiment) under our imaging conditions (Fig 1E and F, and Video EV2). This indicates that, once established, the transverse mode in these large cells is just as robust as the longitudinal pole-to-pole mode in a regular rod-shaped *E. coli* cell.

In order to probe the effect of MinE in the process of symmetry breaking, we engineered a strain that co-expresses sfGFP-MinD and MinE-mKate2 from the endogenous *minDE* genomic locus (see Materials and Methods). In shaped bacteria, MinE-mKate2 proteins oscillate in concert with MinD (Video EV3). After the loss of oscillatory activities of both sfGFP-MinD and MinE-mKate2, no heterogeneous MinE pattern was observed prior to the emergence of MinD patches that dictate the axis of symmetry breaking (Video EV2). This is in agreement with the previous finding that MinE relies on MinD for its recruitment to the membrane (Hu *et al*, 2002).

The observed emergence of Min protein patterns from homogeneous states shows several striking features. First of all, after the early stage of MinD membrane binding, which appears to be rather uniform across the cell, the first patch with enhanced MinD density that forms is neither aligned with the symmetry axes nor does it show a preference for the highly curved polar regions. Secondly, Min patterns converge into a stable pattern within a few oscillation cycles. Emerging patterns align with symmetry axes, and exhibit a preference for the characteristic length range discovered previously (Wu *et al*, 2015b), confirming that the geometry-sensing ability of Min proteins is intrinsic and self-organized. The fast emergence and stabilization of Min protein patterns indicates an intrinsic robustness of Min oscillations and an ability to adjust oscillatory patterns dynamically to changes in cell geometry.

## Analytical and computational approach to probe the geometry-dependent symmetry breaking and pattern selection

The experimental observations described above showed that symmetry breaking in spatially almost-homogeneous states can result in stable oscillation patterns of Min proteins. These spatiotemporal configurations are longitudinal and transverse oscillation patterns whose detailed features are dependent on the geometry of the system, in accordance with our previous study (Wu *et al*, 2015b). We therefore set out to gain a deeper understanding of the mechanisms underlying the phenomenon of multistability and the role of cell geometry in determining, regulating, and guiding the pattern formation process and the ensuing stable spatiotemporal patterns. To this end, we performed a theoretical analysis, building on previous investigations of symmetry breaking induced by the

**Box 1: Symmetry breaking by the Turing instability in cellular geometries**

The initial phase of a "symmetry-breaking" process in a nonlinear, spatially extended system is determined by a *mode-selection mechanism*. Consider an initial steady state of the corresponding well-mixed system that is weakly perturbed spatially, by some spatially white noise, for instance. For the planar geometry considered in textbooks and review articles, the initial state is typically a spatially uniform state (Cross & Hohenberg, 1993; Epstein & Pojman, 1998; Murray, 2003). The spectral decomposition of this state gives equal weight to all Fourier modes and, therefore, sets no bias for a particular mode. A system is referred to as being "Turing unstable" if any spatially non-uniform perturbation of a uniform equilibrium fails to decay (as expected due to diffusion) but instead grows into a patterned state. The collection of growth rates plotted as a function of the wave number of the corresponding Fourier modes is called the dispersion relation, and can be computed by a linear stability analysis. The mode with the fastest growth rate is called the critical mode. It sets the length scale of the initial pattern if there is no other bias for a different mode. Such a bias could, for instance, be provided by a specific initial condition that is non-uniform.

It has been shown recently that, in the context of realistic biological systems, a well-mixed state is generically non-uniform for reaction–diffusion systems based on membrane–cytosol cycling and an NTPase activity (Thalmeier *et al*, 2016). Hence, in this generic case, the symmetry of the stationary state is already broken—in the sense that it is adapted to the geometry of the cell. Consequently, any downstream instabilities—such as the Turing instability—will inherit the symmetry of this spatially non-uniform steady state. In this paper, we discuss how the analysis of the instability of such a non-uniform steady state differs from that of the traditional Turing instabilities of uniform states.

oscillatory Turing instability in bounded geometries (Halatek & Frey, 2012).

The results presented in this Section are based on the observation that the selection of the initial pattern (which does not necessarily coincide with the final pattern) depends on both the Turing instability and the system's geometry. While we focus on the latter aspect in the main text, we review in Box 1 how, more generally, a Turing instability facilitates symmetry breaking in a planar geometry, which may help the reader to understand why the interconnection between geometry and the classical Turing mechanism is crucial.

The non-uniformity of the well-mixed state in cell geometries (as noted in Box 1) is not the only salient difference relative to the classical case of a planar geometry. To perform linear stability analysis on a particular system, a set of Fourier modes must be derived that is specific for the boundary geometry of the system. Hence, both the well-mixed state and the spectrum of Fourier modes are generically geometry-dependent. Only a few geometries are amenable to an analytical treatment. A recent advance was the derivation of eigenfunctions for reaction–diffusion systems with reactive boundaries (the cell membrane) and diffusive bulks (the cytosol) in an elliptical geometry (Halatek & Frey, 2012). This geometry, being analytically accessible, permits broad, systematic parameter studies. At the same time, it shares the symmetries of interest with rod-shaped, circular, and rectangular cells. The eigenfunctions or modes of the ellipse are classified into even and odd functions by their symmetry with respect to reflections through a plane along the long axis; the lowest-order modes are shown in Fig 2A. Even functions are

symmetric, and odd functions are anti-symmetric with respect to long-axis reflection. As such, even functions correspond to longitudinal modes, and odd functions to transverse modes. More subtle than the separation into two symmetry classes, but no less significant, is the strict absence of any homogeneous steady states in elliptical systems undergoing cytosolic nucleotide exchange (Thalmeier *et al*, 2016). This can be understood intuitively from a source–degradation picture: Proteins detach from the membrane and undergo cytosolic ADP-ATP exchange. The concentration of ADP-bound MinD drops with increasing distance from the membrane as the diphosphate is replaced by ATP. This yields cytosolic concentration gradients at the membrane that determine the densities of membrane-bound proteins. In an equilibrium state confined to an elliptical geometry, the cytosolic gradients at the membrane cannot be constant, but will vary along the cell's circumference. Hence, a uniform density at the membrane cannot be a steady state of the system, and instead, the new basal state of the system is defined by the elliptical eigenfunction of the lowest order (Fig 2A). This new steady state takes maximal and minimal values at the cell poles and at midcell, respectively. Note that the spatial variation of the density can be very small and may be very difficult to detect experimentally.

So what is the relevance of such a spatially non-uniform basal state? The answer lies in the nonlinear nature of the system. Nonlinearities are known to amplify weak signals. As discussed in Box 1, the selective amplification of parts of a noise spectrum is at the origin of symmetry breaking. The non-uniformity of the well-mixed basal state implies that a spatially uniform initial condition set in a simulation will first adapt to the symmetry of this basal state, even in the absence of any spatial instability. Only after the basal state has been reached can the growth of (linearly) unstable modes begin. In the present case, the geometry of an ellipse imposes a preferred symmetry on the well-mixed state that resembles the symmetry of a striped oscillation (compare the 0th and 2nd even mode in Fig 2A). Therefore, the initial symmetry adaptation process creates a bias in favor of the 2nd even mode corresponding to striped oscillations, which thus dominates the initial growth of patterns. As shown in Fig 2B, striped oscillations dominate the early phase of pattern formation in a wide variety of cell shapes. In a $6.5 \times 2 \times 1.1 \ \mu m^3$ cell, the oscillatory striped mode persists for about three oscillation cycles before the dynamics switch to pole-to-pole oscillations. By contrast, the oscillatory striped mode persists indefinitely in cells with sizes of $9 \times 2 \times 1.1 \ \mu m^3$ and also $9 \times 5 \times 1 \ \mu m^3$. This latter observation differs from our corresponding experimental results in the same geometry, which had revealed the consistent emergence of a transverse mode after the system had passed through a homogeneous phase (Figs 1D and EV1) [though striped oscillations were also observed in cells of this size (Wu *et al*, 2015b)]. Clearly, letting the computational system evolve from a uniform configuration introduces a bias toward even modes, which should disfavor the selection of transverse patterns. This difference led us to conclude that we needed to characterize in detail the physiological relevance of the bias imposed by the non-uniformity of the well-mixed basal state, that is, its robustness against other types of intracellular heterogeneities. This issue is addressed in the following.

Realistic cellular systems contain many different factors that induce asymmetries and heterogeneities: The cytosol and the membrane are crowded, cell shape is never perfectly symmetrical,

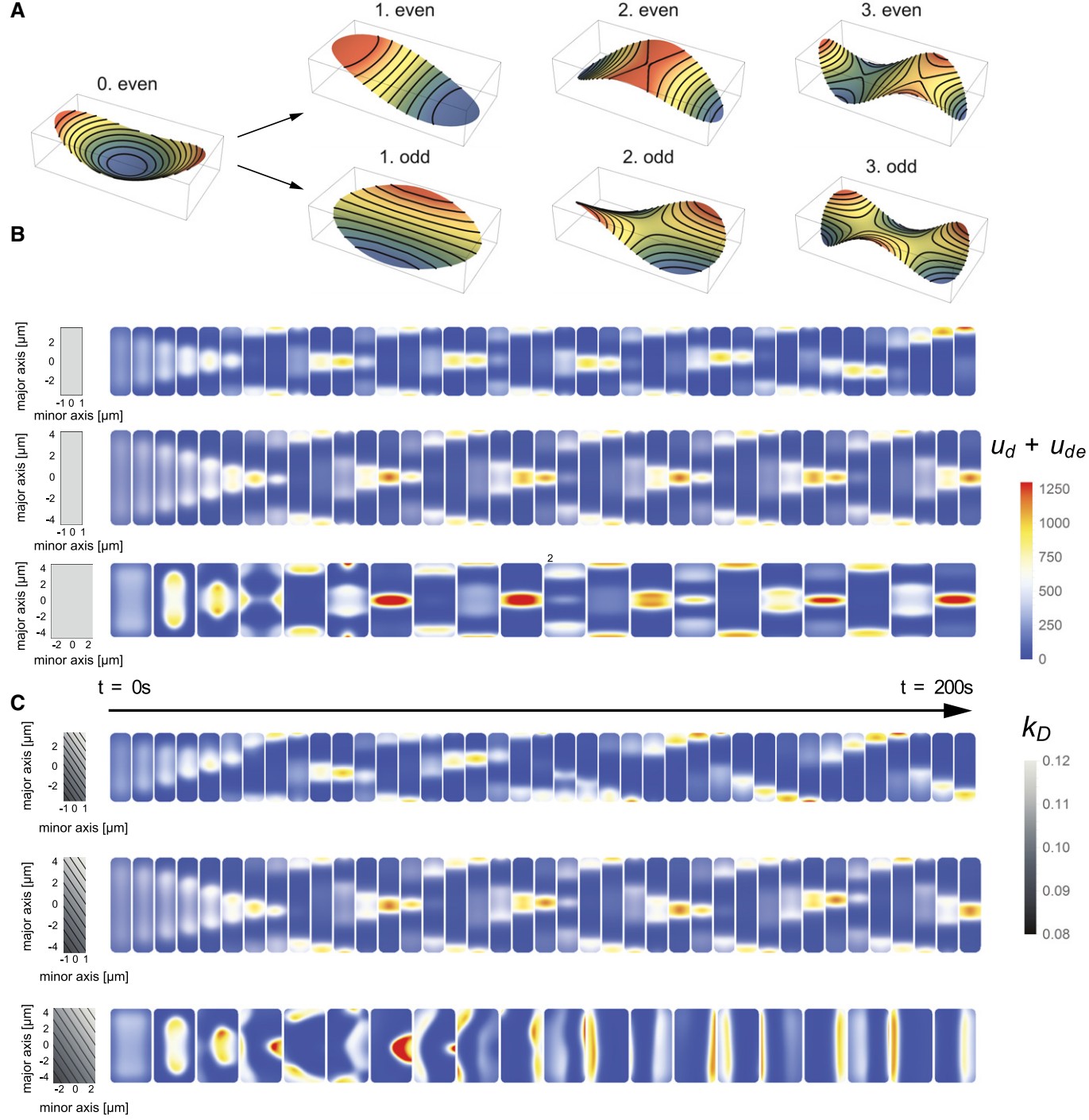

**Figure 2. Pattern emergence upon spatial perturbation.**

A  Even and odd Mathieu functions in an elliptical geometry. The 0.even mode shows the symmetry of the basal state of the system. Here, no homogeneous steady state exists. Note the similarity between the 0[th] and the 2[nd] even mode.

B  Simulations of Min pattern formation from an initially homogeneous state. Dimensions of the cells shown are 6.5 × 2 × 1 μm³, 9 × 2 × 1 μm³, and 9 × 5 × 1 μm³. All cells show an initial striped pattern, which persists in both cells of 9 μm length throughout the simulation period.

C  Simulations analogous to the experiments shown in Fig 1B, with the same cell dimensions as in Fig 2B. The left-hand column depicts the spatially perturbed MinD attachment profile, showing gradients along the diagonal lines of the rectangles. With these attachment profiles, the Min distributions in the three cells quickly evolve into longitudinal, striped, and transverse patterns, respectively.

and the lipid distribution (and hence the membrane's affinity for MinD) is sensitive to membrane curvature. All these intrinsic perturbations of the system's symmetry can have an effect on the

process of pattern selection if multiple stable patterns are possible. Previous studies (Halatek & Frey, 2012) have suggested that *stable* Min patterns are not destabilized by spatial heterogeneities in the

rate of attachment of MinD to the membrane, as the dynamics are dominated by the recruitment process. Here, faced with a multistable system, we asked whether heterogeneities in MinD membrane attachment might to some extent affect the *initial* selection process. To this end, we spatially perturbed the MinD attachment rate by superimposing a linear gradient. We systematically altered the slope and direction of this gradient, and investigated the effects on initial MinD dynamics. After a few oscillation cycles, we turned the perturbation off again and continued the simulation without any induced bias (i.e. with spatially uniform MinD attachment rates). This procedure provided us with a versatile means of generating a weak spatial perturbation that can break symmetry and is applicable to all cell geometries. In particular, it enabled us to quantify the effects of these intrinsic perturbations on pattern selection and compare them to the impact of the geometric bias discussed above.

Indeed, our simulations showed that an initial MinD attachment gradient with a spatial peak-to-peak amplitude of the spatial variation of as little as 20% indeed compensates for the aforementioned geometric bias for striped oscillations (Fig 2C). To put this 20% variation in perspective, we note that the affinity of MinD for different lipids can vary by up to one order of magnitude (Mileykovskaya *et al*, 2003; Renner & Weibel, 2012). Figure 2C shows the onset of pattern formation obtained from computer simulations based on the same geometry as that in Fig 1B. In contrast to the simulations in Fig 2B, the MinD attachment gradient is now initially aligned diagonally. Two observations stand out: Firstly, we find that the asymmetric template does not impede the formation of stripes. Hence, the template does not dictate the symmetry of possible patterns. Secondly, in the 5-μm-wide cells with the weak initial gradient, the transverse mode wins the competition against striped oscillations, which contrasts with the outcome shown in Fig 2B. We accordingly conclude that the geometric bias for striped oscillations is rather weak and is presumably of little physiological relevance. However, in the absence of any intrinsic heterogeneity, pattern selection obtained from computer simulations in cellular geometries will inevitably overemphasize the effect of the geometric bias.

We therefore sought a solution, discussed in the following sections, which explicitly incorporates spatial heterogeneities that compensate for the intrinsic bias, thus effectively restoring unbiased pattern selection based on the Turing instability alone.

**Computing pattern stability in multistable regimes**

Now that we have learned how the initial pattern selection process can be affected by spatial perturbations, we will address how and to what extent the existence and stability of different patterns is affected by the system's geometry, and which molecular processes in the Min reaction circuit control how the system adapts to cell geometry.

Geometry sensing requires the existence of a characteristic length scale. Previous theoretical analysis of Min oscillations has shown that such a length scale is accompanied by synchronization of the depletion and initiation of old and new polar zones, respectively (Halatek & Frey, 2012). A key insight was that a relatively high rate of MinD recruitment (relative to MinE recruitment) is essential for initiation and amplification of the collective redistribution of MinD that leads to such synchronization (Halatek & Frey, 2012). For a

broad range of MinD recruitment rates, we found that oscillatory pole-to-pole and striped oscillations could coexist in cells whose length exceeds a certain limit (Halatek & Frey, 2012; Wu *et al*, 2015b). These earlier studies suggested that the ratio of MinD to MinE recruitment rates is the parameter that allows for geometry-dependent multistability in rectangular cells in which longitudinal and transverse patterns can coexist. The experimental observation of a transverse mode (Wu *et al*, 2015b) supports the previous theoretical suggestion that circular and aberrant patterns in nearly spherical cells (Corbin *et al*, 2002) are caused by the additional destabilization and persistence of odd (transverse) modes in an elliptical geometry with increased cell width (Halatek & Frey, 2012). This implies that the circular and aberrant patterns found experimentally in cells with low aspect ratios, such as nearly spherically shaped cells (Corbin *et al*, 2002), and the observation of transverse patterns in rectangular shapes (Wu *et al*, 2015b), are attributable to the same mechanism, namely the additional destabilization of odd modes. The key difference between the nearly spherical and rectangular cases is that, in the former, the choice of modes is reversible (i.e. neither mode is definitively selected), such that the axis of oscillation switches aberrantly, whereas in rectangular cells the high aspect ratio of the geometry leads to the mutually exclusive selection of either longitudinal (purely even) or transverse (purely odd) patterns, but both symmetries of the pattern are initially accessible (i.e. the system exhibits multistability).

To gain further insight into pattern selection, we first computed and compared the growth rates of even and odd modes in a simplified 2D elliptical geometry, and then proceeded to test the results of this linear stability analysis by computer simulations that take the full 3D cell geometry into account. In these computer simulations, the pattern stability was then probed by the application of spatial heterogeneities in the MinD attachment rate.

As a first step, we performed a linear stability analysis in the elliptical geometry. To characterize the difference between growth rates of even (longitudinal) and odd (transverse) modes, we introduce a quantity which we term the *non-degeneracy*. This is defined as the Euclidian distance between the growth rates of the first three even and the first three odd modes (cf. Materials and Methods section; note that the notion "growth rates of modes" is not to be associated with the physiological growth rates of cells). Figure 3A shows how the non-degeneracy depends on cell geometry and on the MinD recruitment rate. In agreement with our previous analysis, nearly spherical cells are almost degenerate with respect to even and odd modes (Halatek & Frey, 2012). The effect of a larger MinD recruitment rate is to extend this region of near degeneracy toward larger aspect ratios. Hence, when rates of MinD recruitment are high, we can expect that longitudinal and transverse modes have similar growth rates even in rectangular cells. These results were then tested in 3D computer simulations.

For simulations of realistic 3D cellular geometries, we employ a spatially varying MinD attachment rate, similar to the approach described in Section Analytical and computational approach to probe the geometry-dependent symmetry breaking and pattern selection. This allows us to probe the stability of patterns against spatial perturbations, and thereby to test the (nonlinear) stability of the oscillatory pattern. The simulation strategy is schematically shown in Fig 3B. First, we align the gradient of the MinD attachment profile with one symmetry axis and initialize the simulation.

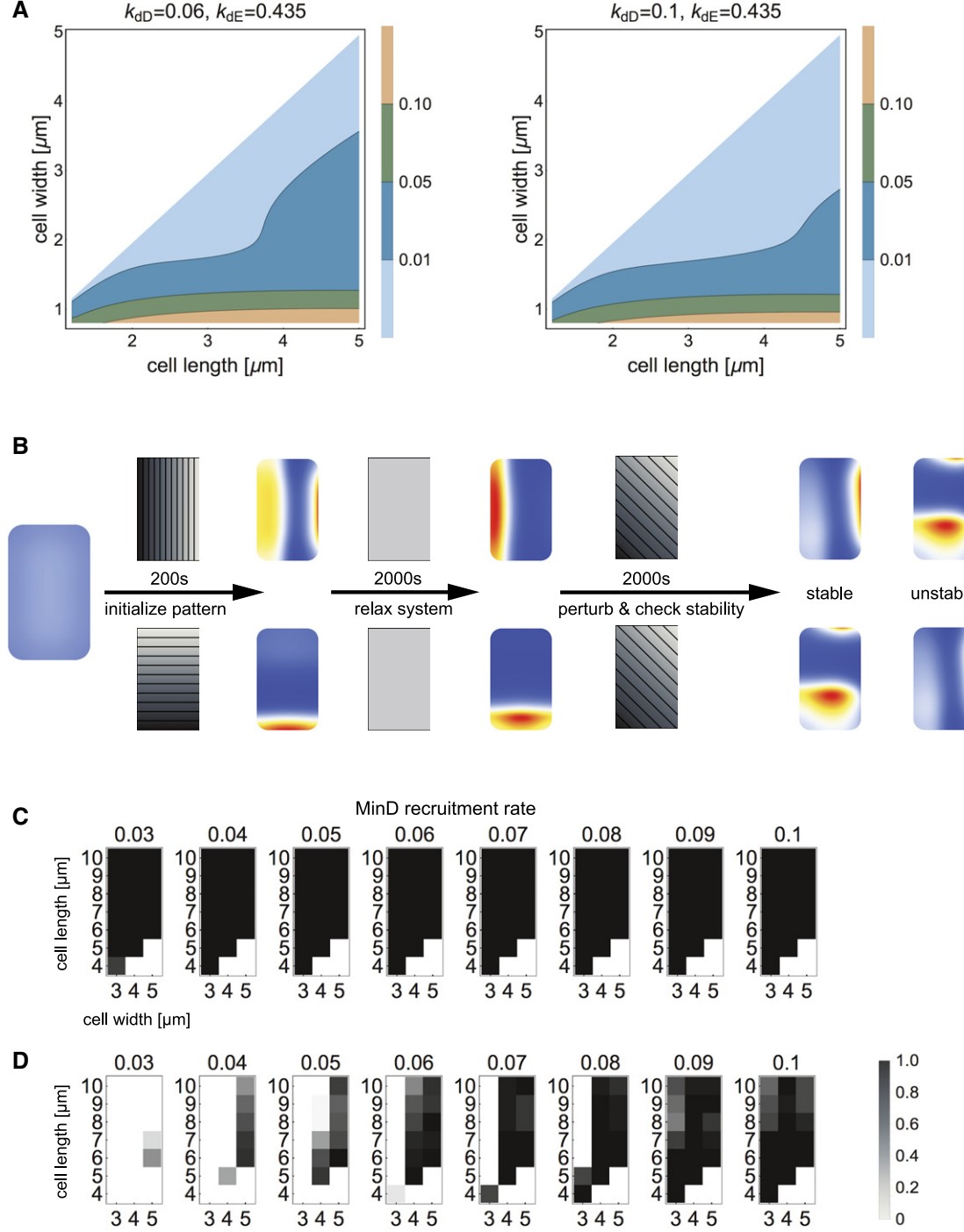

**Figure 3. Computing stability in multistability regimes.**

A    Two plots that show the non-degeneracy of even and odd modes in an elliptical geometry for varying cell geometry and MinD recruitment rate (*kdD*). The unvaried MinE recruitment rate is noted as *kdE*. The degeneracy (light blue area) increases with the MinD recruitment rate.

B    Schematic representation of the simulation process used to probe the stability of longitudinal and transverse patterns. The system is initialized with a homogeneous configuration, and the gradient of the MinD attachment rate is aligned with the major or minor axis to direct pattern selection. After initialization, the MinD attachment rate is equalized to allow the system to relax into the initialized state. If the initialized pattern persists in the absence of a stabilizing gradient, the gradient is reapplied to deflect the pattern from its preset alignment and study its stability *vis-a-vis* spatial inhomogeneities that break its symmetry. The stability toward all possible deflections with linear MinD attachment profiles is probed, and the persistence of the initialized pattern is checked.

C, D  Stability diagrams of the simulation procedure outlined in (B) for longitudinal (C) and transverse (D) patterns. White areas represent configurations where the respective mode was not initialized. The gray values show the fraction of all simulations (with different attachment templates) in which the respective pattern mode is sustained.

After a few oscillation cycles, we turn the MinD attachment gradient off and allow the simulation to proceed for another ~40 oscillation cycles. If the pattern was stable (i.e. a local attractor of the reaction–diffusion dynamics), it remained aligned with the initially selected axis. In these cases, we used the final state as the initial configuration and ran the simulation for another ~40 oscillation cycles, now with reactivated perturbation of the MinD attachment rate and with the gradient inclined at an angle to the initial oscillation axis. This final step was intended to probe the stability of the pattern against spatial heterogeneities that could potentially switch pattern symmetry from longitudinal to transverse or vice versa. We repeated this simulation to cover all possible alignments (i.e. angles from 0 to 90 degrees) and slopes of the MinD attachment perturbation (i.e. spatial variations from 0 to 100% of the average MinD attachment rate). Together, these simulations enabled us to quantify the stability of each initialized pattern based on the degree of perturbation that it can sustain without losing its alignment to the initial axis. We performed this stability analysis for a broad range of experimentally probed geometries as well as recruitment rates. Note that we only distinguished transverse oscillations from longitudinal oscillations, but not between pole-to-pole and striped modes within the longitudinal oscillations. In all probed configurations (cell geometries, spatial heterogeneities), we observed that longitudinal patterns are stable, independently of the MinD recruitment rate (Fig 3C). In contrast, the number of cell geometries that support stable transverse patterns turned out to be strongly dependent on the relative rate of MinD recruitment (Fig 3D). In agreement with the above linear stability analysis in the 2D elliptical geometry, we found that an increasing MinD recruitment rate extends the domain of stable transverse patterns toward cell geometries with larger aspect ratios. Furthermore, our simulations show that the degree of pattern stability is surprisingly high. Almost all configurations were able to withstand more than 90% of all applied perturbations (slopes and angles) to the MinD attachment profile (Fig 3C and D).

These findings lead to several important conclusions. First, the simulation data show that stability analysis in the two-dimensional elliptical geometry is able to account well for the patterns of behavior observed in realistic three-dimensional geometries. Second, our findings indicate that a gradient in the MinD attachment rate affects the initial selection of the axis of oscillation by guiding the dynamics into the basin of attraction of the corresponding pattern. Moreover, spatial gradients of MinD attachment rate typically cannot drive a system from one pattern into the orthogonal alternative once the system has settled down into a stable oscillation. This suggests that the spatiotemporal patterns are in general very robust against spatial heterogeneities in the MinD attachment rate. The above analysis provides a way to probe the basins of attraction of different oscillatory patterns systematically, which will be introduced and discussed in the following.

## Basins of patterns are controlled by geometry and recruitment strength

In the preceding Section, we demonstrated that highly stable longitudinal and transverse patterns can be initialized in a broad range of geometric configurations. Knowing that these patterns exist, we turned to the question of which patterns can be plausibly reached by the system dynamics, that is, without having to tune the initial

conditions in any particular fashion. To approach this issue, we began our simulations with a homogeneous initial configuration. As discussed in Section Analytical and computational approach to probe the geometry-dependent symmetry breaking and pattern selection, adaptation to the non-uniform well-mixed state (adaptation to geometry) introduces a preference for striped oscillations, and hence a bias for even patterns. To include other potential effects that weakly break the system's symmetry (but not the symmetry of the stable patterns, cf. Section Computing pattern stability in multistable regimes) and neutralize the weak bias for striped selection, we imposed a fixed, weak spatial gradient on the rate of MinD attachment. The relative magnitude of the variation was again set to 20%, which, as mentioned above, is well below the typical range of variation in MinD's affinity for different lipids in the *E. coli* membrane. We examined all alignments of the MinD attachment gradient interpolating between purely longitudinal and purely transversal states. After ~100 oscillation cycles, we recorded the final pattern, distinguishing between transverse pole-to-pole, longitudinal pole-to-pole, and longitudinal striped oscillations. Following this procedure, we separately studied the effects of varying geometry and MinD recruitment rates on multistability and pattern selection.

To study the effect of system geometry, we fixed the value of the MinD recruitment rate to a high value ($k_{dD} = 0.1$) such that the number of coexisting stable longitudinal and transverse patterns is largest. Sampling over all alignments of the gradient led to the distributions of the final patterns shown in the histograms in Fig 4A. Cell length was varied from 7 to 10 μm, cell width from 3 to 5 μm. We observed a critical cell length of between 9 and 10 μm for the selection of striped oscillations. This coincides with the length scale for which the model parameters were initially adjusted in the 2D elliptical geometry (Halatek & Frey, 2012). Surprisingly, this length scale translates directly to realistic 3D cell shapes. We found that the fraction of oscillatory striped patterns decreased in favor of transverse patterns as the cell width was increased. Overall, these results show that cell width, and not cell length, is the main determinant for the onset of transverse modes. All these observations are remarkably consistent with previous experimental data based on random sampling of live *E. coli* cells after they have reached a defined shape (Wu *et al*, 2015b). Given this level of agreement, we expected to gain further insight into the molecular origin of the observed pattern distribution by studying its dependence on the kinetic parameters in the theoretical model.

To investigate the effect of MinD recruitment rate, we focused on data from the cell sizes that show the greatest number of coexisting patterns, as determined by the previous numerical stability analysis. The corresponding histograms are shown in Fig 4B. The cell lengths for which the data was collected were 9 and 10 μm, and the cell width varied from 1.1 to 5 μm. In narrow cells, we recovered our previous results on the onset of striped oscillations: The fraction of stripes increased with the MinD recruitment rate (Halatek & Frey, 2012). Remarkably, this was no longer the case when cells reached a width of 5 μm: Here, the fraction of stripes was zero below some threshold MinD recruitment rate, and took on a constant value above this threshold. On the other hand, the fraction of transverse patterns did increase with MinD recruitment rate in these 5-μm-wide cells, as does that of the striped fraction in narrower cells. Hence, we conclude that multistability is indeed promoted by high rates of MinD recruitment. We attribute this feature to the ability of the

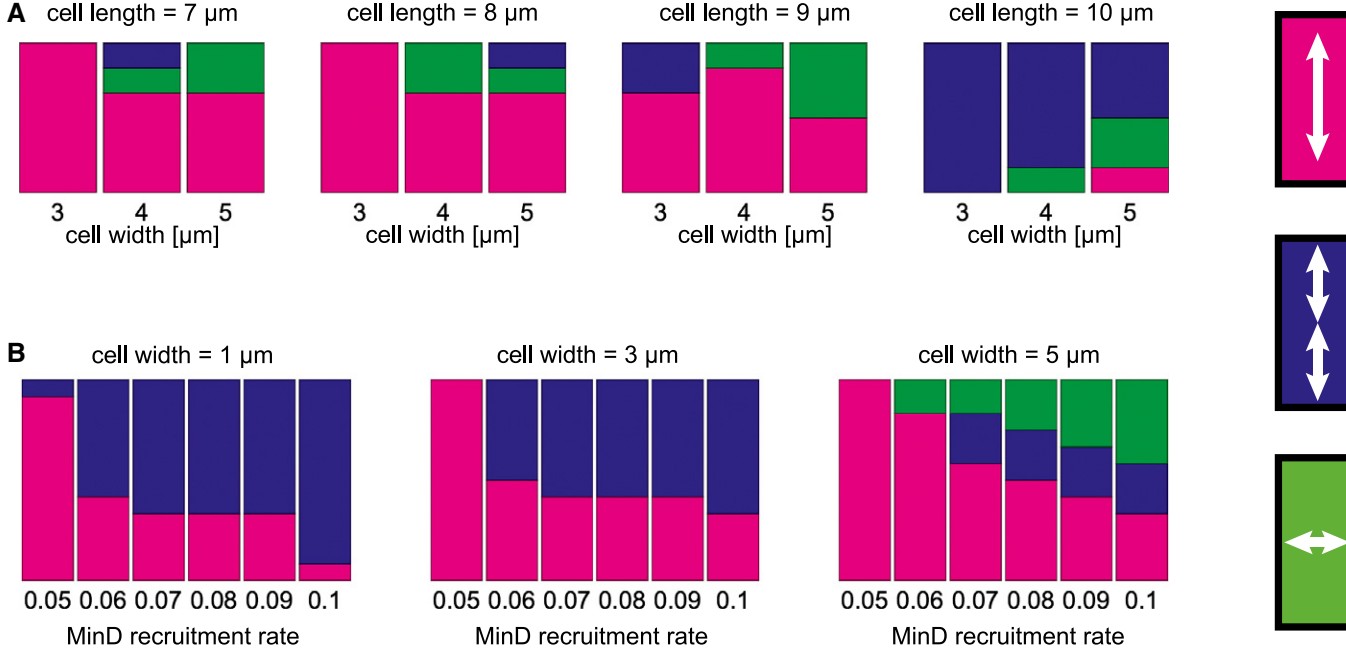

**Figure 4.  Basins of attraction predicted from systematic perturbations of patterns with shallow attachment gradients.**

A   Relative distribution of the final patterns (indicated on the right) observed after sampling all alignment angles of the MinD attachment template from 0 to 90 degrees. The MinD recruitment rate was set to a constant value $k_{dD}$ = 0.1. The data show the increase in the incidence of multistability as the cell size is increased beyond minimal values for cell length and cell width.

B   Fractions of the final patterns in cells of 9 and 10 μm length after sampling all alignment angles of the MinD attachment template from 0 to 90 degrees. The data show that increasing the MinD recruitment rate facilitates multistability.

reaction–diffusion system to operate in the regime in which a characteristic length scale is established through synchronized growth and depletion of spatially separated polar zones ("canalized transfer") (Halatek & Frey, 2012). Notably, the same mechanism that enables striped oscillations in filamentous cells also facilitates transverse oscillations in wide cells.

In all examples discussed so far, the height of the cell was fixed at 1.1 μm, well below the minimal span required to establish a Min oscillation (Halatek & Frey, 2012). Therefore, no oscillations occur along the *z*-axis. While the present study focuses on competition between longitudinal and transverse patterns, we also used our computational model to explore patterns along the *z*-direction. In a representative simulation with a 3.1 μm high chamber (cell dimensions 5 × 4 × 3.1 μm$^3$), we found oscillations aligned with the *z*-axis in addition to oscillations aligned with the *x*- and *y*-axes. This shows that increased headroom in the third dimension extends the number of accessible stable patterns even further.

**Persistent directionality traps Min oscillations in a stable state during cell growth**

Experiments (Fig 1B–F) and simulations have shown that both longitudinal and transverse modes are stable over a range of rectangular shapes once they have been established. However, it is still unclear how patterns evolve during cell growth, which can involve an increase in volume of over 10-fold. Particularly intriguing is the fact that different patterns emerge during the growth of cells that reach the same final shape. This prompted us to study the

development of patterns throughout the growth history of a cell. We captured around 200 successive MinD binding patterns per cell at intervals of 2 min during the geometrical changes that accompanied cell growth. Here, we focused on the cells that reach a final width of between 5 and 5.5 μm and a final length of 8–10 μm, taking advantage of their very long growth history of 6–8 h and the previously detected coexistence of two longitudinal modes and a transverse mode in such cells. The final data set comprised 97 cells.

Spatially constrained by microchambers, the cells adopted growth patterns that can be categorized into several types, based on the difference in alignment of the cell axes with the axes of the chambers (Fig 5A, D and G). Under the combined effects of exposure to A22 and cephalexin, cells are initially elliptical in shape (Fig 5A and D). When cell widths were small, Min oscillations almost exclusively aligned along the longest elliptical axis of the cell, with a certain degree of lateral-axis fluctuation (Fig 5B and E). As a result, with respect to the rectangular chamber axes, the initial Min patterns were aligned in accordance with the orientations of the cells. Figure 5A and D, for example, show two cells whose long axes are initially aligned with the long axis and short axis of the chambers, respectively. In Fig 5B, Min oscillations remained aligned close to the vertical (long) axis for the entire 7.8 h of cell growth, from an initial size of 2.1 × 1.5 × 1 μm$^3$ (at *t* = 0) to a final size of 9 × 5 × 1 μm$^3$ (Fig 5A; for other examples see Video EV4). In contrast, Min oscillations in Fig 5E aligned close to the horizontal (short) axis of the chamber over the whole 8 h taken to reach the same dimensions (Fig 5D; for more examples see Video EV4). Note that in the latter scenario, the long and short axes exchanged

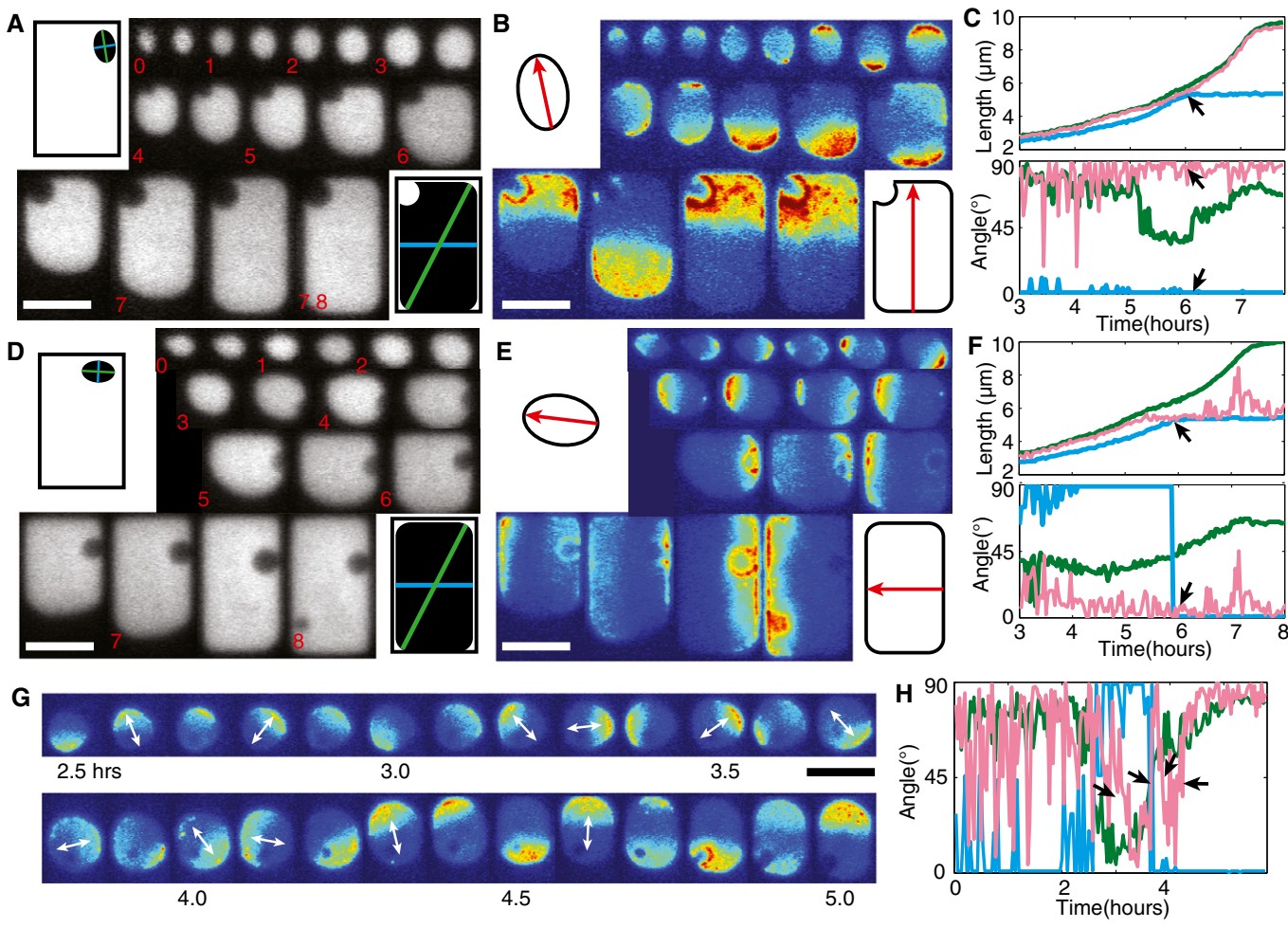

**Figure 5. The effect of cell-shape change during growth on the stability of Min protein patterns.**

A Cytosolic fluorescence during growth of a cell from a small elliptical form into a large rectangular shape. Numbers in red indicate time in hours. Illustrations show the positions and orientations of the cell in the first and last time frames. Green and blue lines indicate the maximum and minimum Feret diameters, respectively.

B sfGFP-MinD patterns during the growth of the cell shown in (A). Illustrations indicate the cell boundaries and oscillation angles observed in the first and last frames (not to scale).

C Quantitative data obtained from the cell shown in (A and B). The maximum and minimum Feret diameters (green and blue), and the measured MinD oscillations (red) were expressed in terms of length (top) and angle (bottom) and plotted against time. The number of cells that fit this category was 41/97. Arrows indicate the time when cell width reached the chamber width of 5 μm.

D–F Data are presented as in (A–C) for another cell that showed persistent oscillations along the horizontal axis throughout growth. The number of cells that fit this category was 28/97.

G Time-lapse images of sfGFP-MinD that reveal stochastic switching of patterns in a cell with an asymmetric shape and a low aspect ratio. White arrows indicate the oscillation axes.

H The angles of the maximum and minimum Feret diameters (green and blue), and the measured MinD clusters (red) for the cell shown in panel (G) plotted against time. The number of cells that fit this category was 10/97.

Data information: All scale bars correspond to 5 μm.

identity at *t* = 5.8 h, but this did not affect the persistence of horizontal Min oscillations (Fig 5D and E). These observations suggest that Min oscillations have a strong tendency to remain faithful to their existing orientation for as long as the length scale allows. In addition, some pattern transitions were observed during instances of drastic switching of cell axes that are associated with a low aspect ratio of the cell shapes (Fig 5G and Video EV5), similar to examples shown previously (Corbin *et al*, 2002; Männik *et al*, 2012). This phenomenon was explained previously by invoking theoretical predictions that low aspect ratios should lead to a transient coupling

between longitudinal and transverse modes (Halatek & Frey, 2012) and Min patterns in these shapes are more sensitive to stochastic perturbations (Fange & Elf, 2006; Schulte *et al*, 2015). The above scenarios show that pattern multistability can emerge through adaptation of persistent Min oscillations during different modes of cell growth.

To quantitatively characterize the evolution of Min patterns during cell growth, we wrote a data analysis program that automatically quantifies cell shape and Min patterns (see Materials and Methods, Fig EV2). We used Feret's statistical diameters to parameterize

cell shape. Feret's diameter measures the perpendicular distance between two parallel tangents touching the opposite sides of the shape (Walton, 1948). This can be measured along all angles, and the maximum and minimum values are used here to define the smallest and largest cell dimensions. In general, the minimum Feret diameter aligns with the short (symmetry) axis of the cell; the maximum Feret diameter aligns with the long axis of a near-elliptical shape and the diagonal of a near-rectangular shape. We defined the angle of oscillations by connecting the center of the MinD patch to the cell center. Note that all angles were calculated relative to the horizontal plane. With these measurements, we can now compare the length scale that Min oscillations adopt with the lengths of the cell's dimensions (top panels in Fig 5C, F and H). We can also correlate the angle of the Min oscillations with the planes along which these cell dimensions are measured (bottom panels in Fig 5C and F). Indeed, Fig 5C and F show that Min patterns aligned with either the long (symmetry) axis or the short (symmetry) axis of the cell shapes, albeit with some degree of fluctuation. In addition, the frequent switching of Min oscillation angles in cells with low aspect ratios is well captured by the automated analysis (Fig 5H).

For statistical analyses of the robustness of Min oscillations against cell-axis switching, we evaluated Min patterns 20 min before and 20 min after the time point at which cell width reaches the limit of 5 μm imposed by the width of the chamber (marked by the black arrows in all plots in Fig 5C and F). At the beginning of this period, all Min patterns were in longitudinal pole-to-pole mode. Over the following 40 min, 41 of the 97 cells analyzed showed no large-scale axis shift, with the long axes remaining above 75° and the short axes below 15°. In all these cells, Min oscillations were sustained along the vertical (long) axes, as shown in Fig 5A–C. Maintenance of the oscillations along the long axis was also observed in 18 cells in which the long axis did not undergo a drastic switch but the short axis did. In total, 60% of the cells exhibited continuous alignment with the long axis during adaptation of the cell to the width of the chamber. The other 40% of the cells showed a switch in the mode of oscillations, including 28 cells that followed a similar pattern of growth to those shown in Fig 5D–F and 10 cells that grew as in Fig 5G and H.

These observations reveal several features. First of all, a robust long-axis alignment of Min patterns in narrow cells determines the initial oscillation direction. Second, the directions of established oscillations are sustained for as long as the corresponding cell dimension along this direction falls within the characteristic symmetry and scale preferred by the oscillation mode (e.g. a 5-μm horizontal dimension in Fig 5D). Third, Min oscillations show a notable degree of tolerance to asymmetries in cell shape during growth. These properties largely agree with our previous conclusion that the propensity to adopt a given pattern is set by the length scale and the symmetry of the cell shape (Wu *et al*, 2015b). Hence, in a cell shape that allows for multistability, the selection of Min pattern mode depends largely on (and thus is deducible from) the growth history of the cell.

## Experimental observations of pattern transitions between multistable states

In large cells, 5 μm in width, we observed transitions from longitudinal pole-to-pole modes to transverse modes and vice versa (Fig 6A

and B, and Video EV6). These transitions occurred after the long and short axes of the cell had aligned with the respective axes of the chambers due to confinement, and were characteristically different from the transitions caused by low aspect ratio and shape asymmetry shown in Fig 5G. For instance, Fig 6A shows a transition from the longitudinal to the transverse mode. This transition initiated with a large and unexpected displacement of the MinD polar zone from the longitudinal axis of the cell ($9 \times 5 \times 1$ μm$^3$) after several hours of persistent longitudinal oscillations. This perturbation gradually shifted the axis of oscillation toward the short axis of the cell over the course of 10 oscillation cycles. An example of the inverse transition is shown in Fig 6B for a $6 \times 5 \times 1$ μm$^3$ cell. We note here that this type of spontaneous rearrangement of the oscillation mode occurred rather infrequently, considering the 6- to 8-h lifetime of a bacterium on the chip. To distinguish this type of transition from the previously discussed transitions induced by small aspect ratio or apparent asymmetry (cf. Fig 4H), we restricted the further statistical analysis to data from the growth phase after the point at which the maximum cell width of 5 μm had been attained. This phase spanned the last 2–3 h of cell growth, that is, encompassed 120–180 min oscillation cycles. We found that the majority of cells that eventually came to occupy a volume of $9 \times 5 \times 1$ μm$^3$ ($n = 47$, excluding the few cells that went through a transient homogeneous state such as that shown in Fig 1B) only exhibited one transition in their Min patterns (Fig 6C). Transitions rarely occurred more than once in any given cell. On average, 0.3 transitions occurred per cell per hour during growth from a size of $6 \times 5 \times 1$ μm$^3$ to a size of $9 \times 5 \times 1$ μm$^3$, and this observation holds true for cells grown in both nutrient-rich and nutrient-poor media (see Materials and Methods). The average number of transitions per cell did however increase in nutrient-poor medium (see Fig 6C, inset), which correlates well with the fact that it took them longer to fill out the custom-designed shapes. Altogether, the rarity of such transitions again confirms that different pattern modes are robust against intracellular fluctuations.

Automatic angle tracking of the sfGFP-MinD clusters reveals that most of the transitions between longitudinal and transverse modes involve an intermediate state in which the axis of oscillation deviates from the symmetry axes of the cell shape (Fig 6D). This suggests that the transitions are due to a strong perturbation of a stable oscillation that pushes the system into the domain of attraction of another stable oscillatory mode. Most of these gradual transitions took place on time scales of 4–8 min in both nutrient-rich and nutrient-poor growth medium (Fig 6E and inset).

The types of transitions occurring in these cells are length dependent (Fig 6F). In our data set, transitions from transverse to longitudinal mode were only found in cells with lengths around 6 and 7 μm, whereas the inverse transition was only observed at cell lengths of around 8–9 μm. In such cells, the longitudinal striped oscillation mode was observed to evolve from either longitudinal or transverse pole-to-pole oscillations at lower frequencies.

To explore the effect of cell width on pattern stability, we carried out long-term time-lapse imaging of cells shaped into rectangles with lengths of 9–10 μm and widths of 3–6 μm (Fig 6G). Unlike previous experiments, in which we had randomly sampled cells that had already attained the desired shape and imaged them at 2-min intervals (Wu *et al*, 2015b), here we were able to determine the final pattern before cell death or before cells grew out of

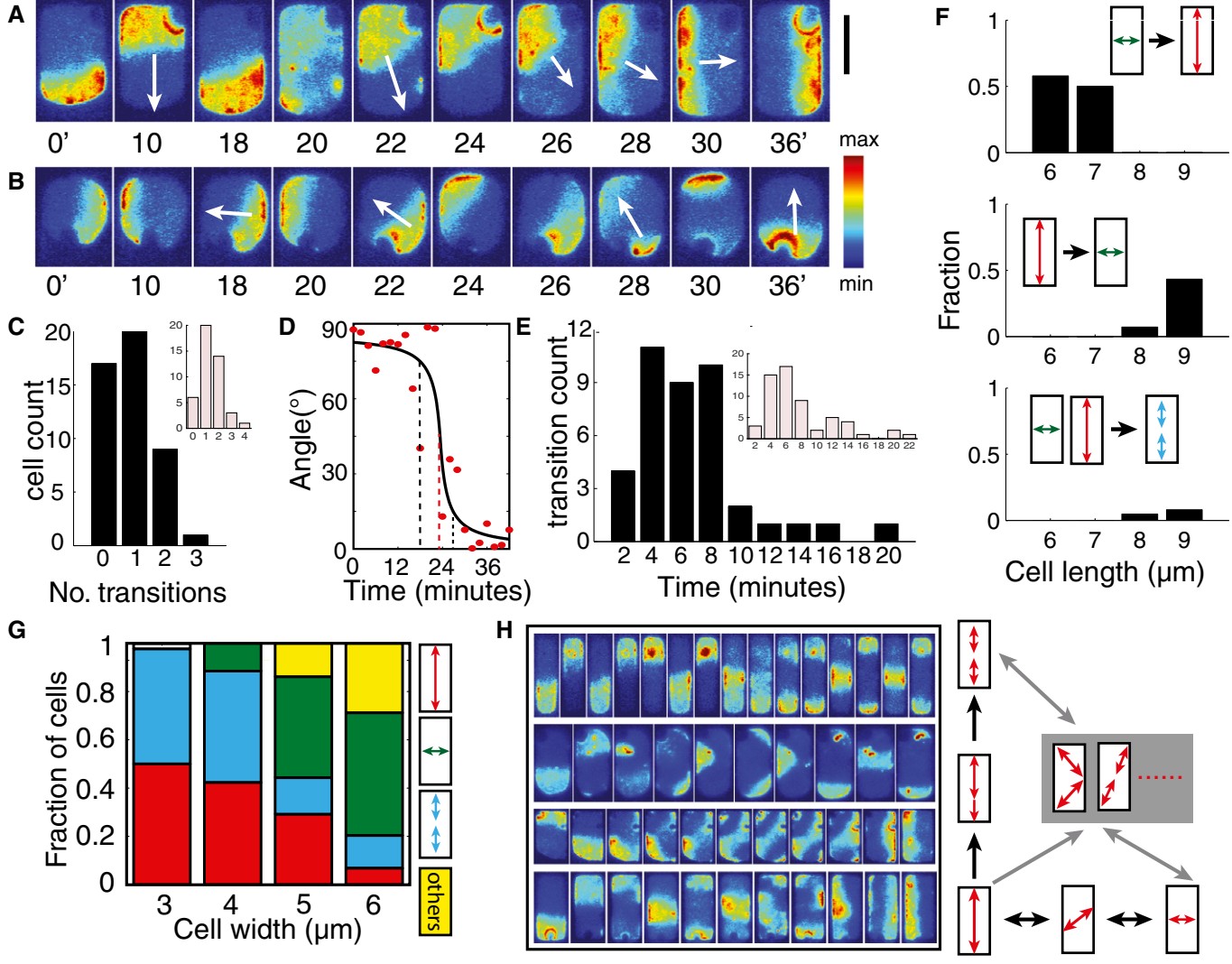

**Figure 6. Transitions between various modes of Min protein patterns.**

A    Time-lapse images showing the transition from longitudinal pole-to-pole mode to transverse mode. Scale bar, 5 μm.

B    Time-lapse images showing the transition from transverse mode to longitudinal pole-to-pole mode.

C    Bar plot showing the distribution of the number of transitions. Inset: Data from experiments carried out under nutrient-poor conditions in which growth rates are reduced.

D    Representative time course of a change in the mode of sfGFP-MinD oscillation. The black line is a sigmoidal fit. The dashed black lines indicate 15° and 75°, and the dashed red line indicates 45°.

E    Bar plot showing the time scale of the switch in the oscillations. Inset: Data from experiments carried out in nutrient-poor conditions.

F    Bar plots showing the relative numbers of the indicated transitions that occur at different cell lengths. All cells have a width of 5 μm.

G    Distribution of final patterns in cells of the indicated widths . Cell lengths are all 9–10 μm.

H    Time-lapse images of various modes of transitions between patterns. Cell sizes from top to bottom are, respectively, 10 × 2 × 1, 10 × 6 × 1, 9 × 5 × 1, 10 × 4 × 1 μm³. Note that the cells are scaled differently. On the right is an illustration showing Min pattern transitions through intermediate states.

the chamber. In agreement with the trend seen in previous experiments, increase in cell width resulted in a reduction of the fraction of cells displaying oscillations in the longitudinal pole-to-pole mode and a corresponding increase in the proportion of the transverse mode. Strikingly, we find that the incidence of oscillatory striped patterns decreases dramatically as cell width increases from 4 to 5 μm. This feature was also well captured by the simulation data in Fig 4A. Hence, while the precise pattern mode in a cell depends on various factors including growth history and large

intracellular perturbations, the statistical trend in pattern composition with respect to cell size is compatible with the basins of attractions probed through small spatial perturbations in our simulations (Fig 4A).

When cell widths reached more than 5 μm, more complex oscillation modes were observed, including diagonally striped, zigzag and other asymmetric patterns. These modes often appeared to represent transient, intermediate states between two symmetric modes (Fig 6H and Video EV6), but could occasionally persist for

several cycles before cell death or overgrowth, as presented in the statistics in Fig 6G. Thus, increasing cell width expands the number of intermediary metastable states available for transitions between stable oscillation modes (Fig 6H). In addition, a transverse–striped mode has also been observed (albeit infrequently) in cells with widths of slightly over 6 μm (Video EV6), further demonstrating that the 3- to 6-μm adaptive range dictates mode selection in Min pattern formation.

## Discussion

Combining experiments and theory to study the time evolution of Min oscillations in shaped bacteria, this work sheds new light on the origin of multistability in biological Turing patterns and on transitions between different patterned states. The experiments described here show how a stable pattern can emerge from a homogeneous state via direct symmetry breaking. Moreover, these patterns exhibit persistent adaptation during cell growth, as well as dynamic transitions induced by strong spatial perturbations. Systematic stability analyses of multistable states *in silico* revealed that the underlying Min pattern dynamics is set by (i) the sensitivity of initial pattern selection to cellular heterogeneity and (ii) the robustness of the established oscillations in the face of perturbations. Overall, this study establishes a framework for understanding Turing reaction–diffusion patterns in the context of fluctuating cellular environments and boundary growth.

Any study on the emergence of patterns within a cellular boundary must take cellular heterogeneity into account. Homogeneous initial states have been broadly used to probe the emergence of spatial patterns in computational simulations. While such an approach has been shown to capture the symmetry breaking of unbounded reaction–diffusion systems, we demonstrate that computing pattern selection in *bounded* systems from such a homogeneous initial state can lead to an intrinsic (but physiologically irrelevant) bias. For example, in this study, a bias toward striped modes impedes computer simulations that employ a homogeneous initial state from reaching a transverse pattern, even if the stability of such a transverse pattern is comparable to that of a longitudinal pattern. The new theoretical methods outlined in this study provide a framework for realistically predicting symmetry breaking in biological systems through linear stability analysis in an elliptical geometry, and probing the basins of attraction of different stable patterns by numerical simulations. Our examples demonstrate the importance of taking spatial heterogeneity into account when studying symmetry breaking within biological boundaries.

Multistability in Min patterns is not determined by either kinetic parameters or cell geometry alone, but originates from the interdependence between the geometric properties of the cell's form and the kinetic regimes of the pattern-forming system. Some limited examples of multistability in reaction–diffusion systems have previously been analyzed in very large systems (Ouyang *et al*, 1992), where the system size exceeded the length scale of the pattern by two orders of magnitude and the system geometry was rotationally symmetrical. Here, the various stable states of Min patterns are defined with reference to the axes of cell shape, and boundary confinement is thus required by definition, without being a sufficient condition, for the emergence of the class of multistability

phenomenon characterized in this study. For instance, an increase of cell width beyond 3 μm is required to enable the transverse mode to be sustained in addition to a longitudinal pole-to-pole oscillation. Most interestingly, our theoretical analysis of the underlying model shows that increasing the size of a Turing-unstable system alone does not in itself facilitate the existence of multiple stable patterns that can be reached from a broad range of initial conditions. In our previous theoretical work, we had found that the emergence of a pole-to-pole oscillation in a short cell does not generically imply the existence of a stable striped oscillation with a characteristic wavelength in a long filamentous cell (Halatek & Frey, 2012). Instead, the emergence of a characteristic length scale (which becomes manifest in striped oscillations) is restricted to a specific regime of kinetic parameters, where growth and depletion of spatially separated polar zones become synchronized such that multiple, spatially separated polar zones can be maintained simultaneously. A key element among the prerequisites that permit this regime to be reached is that the nonlinear kinetics of the system (MinD cooperativity) must be particularly strong. Here, we find the same restriction on the parameters for the emergence and selection of stable transverse patterns in addition to longitudinal pole-to-pole and striped oscillations. For example, weak nonlinear (cooperative) kinetics can readily give rise to longitudinal Min oscillations in 2-μm-long cells, but cannot sustain a transverse mode of oscillation in cells as wide as 4 μm. These findings hint at an exciting connection between multistability, the ability of patterns to sense and adapt to changes in system geometry, and the existence of an intrinsic length scale in the underlying reaction–diffusion dynamics. Remarkably—and contrary to the treatments in the classical literature—the existence of an intrinsic length scale is not generic for a Turing instability *per se*. One example is the aforementioned selection of pole-to-pole patterns in arbitrarily long cells where MinD recruitment is weak. In this case, irrespective of the critical wave number of the Turing instability, the final pattern is always a single wave traveling from pole to pole. The selection of a single polar zone is also characteristic in the context of cell polarity (Otsuji *et al*, 2007; Klünder *et al*, 2013), where it has been ascribed to the finite protein reservoir and a winner-takes-all mechanism. It will be an interesting task for further research to elucidate the general requirements for the emergence of an intrinsic length scale in mass-conserved reaction–diffusion systems. Here, we have defined the requirements for geometry sensing and multistabilty in the underlying model for Min protein dynamics.

The dynamic relationship between multistable states is determined by the robustness of individual stable states when exposed to large-scale intracellular fluctuations. Our computer simulations suggest that the Min system can tolerate various degrees of spatial perturbations imposed by a heterogeneous profile of MinD's binding affinity for the membrane. This is consistent with our experimental observation that a Min oscillation mode can persist in a living cell for tens of oscillation cycles, even within cell shapes where other stable states exist. Such persistence was also found to tolerate a large degree of asymmetry in cell shape, except for cases with low aspect ratios. Multistable states in the Min system are in essence independent stable states that do not toggle back and forth except under the influence of large spatial perturbations. This is experimentally verified by our observation that instances of switching between multistable states are extremely rare in large rectangular cells.

These properties show that biological patterns driven by a reaction–diffusion mechanism can exhibit behaviors similar to classical bistable systems, in which two states switch from one to the other upon surmounting an activation energy barrier.

Pattern selection among multistable states can be dependent on cell growth. Turing patterns have rarely been analyzed in the context of growth, either experimentally or computationally, largely due to technical challenges. A recent example is the study of digit formation during embryonic development (Raspopovic *et al*, 2014), where a 3-node Turing network was simulated in a 2D growing mesh to verify experimental findings. In the present paper, our study of the Min oscillations throughout the growth history of the cells revealed a remarkable persistence in the face of boundary changes induced by cell growth. This phenomenon could not be deduced from previous studies on the Min system, which showed various degrees of fluctuations in cells with certain degrees of asymmetry and enlargement (Corbin *et al*, 2002; Fange & Elf, 2006; Varma *et al*, 2008; Männik *et al*, 2012; Hoffmann & Schwarz, 2014; Schulte *et al*, 2015). Indeed, although Min oscillations do fluctuate in our experimental settings, they rarely undergo drastic switches even during periods of growth that increase the cell volume by up to 20-fold. One essential finding of this study is the persistent directionality of the oscillations in the case where the long axis and short axis of a cell have switched during adaptation to the chamber boundaries. This provides strong evidence that the Min patterns do not respond to boundary changes *per se*, but are dictated by the history and the scale of the cell dimensions. With such a strong tolerance for physiological and geometrical fluctuations, the patterns are thus found to be largely predictable when the growth history of the cell is known, even without explicit computer simulations involving stochastic effects and boundary growth.

Nonlinear kinetics and boundary confinement are general to Turing patterns in cells and organisms (Vicker, 2002; Goryachev & Pokhilko, 2008; Kondo & Miura, 2010; Klünder *et al*, 2013; Raspopovic *et al*, 2014), implying that the multistability phenomenon can be probed in other reaction–diffusion systems as well. Using the framework employed in this study to understand the effect of fluctuations and growth in these other systems may facilitate the discovery of general rules governing the spatial adaptation of patterns in biology.

# Materials and Methods

### Bacterial strains

In this study, all MinD and MinE proteins or their fluorescent fusions were expressed from the endogenous genomic *minDE* locus. Bacterial strain BN1590 (W3110 [*ΔleuB :: eqFP670 :: frt aph frt, ΔminDE :: sfGFP-minD minE :: frt*]), constructed and characterized previously (Wu *et al*, 2015a,b), was used for all the experiments in this study, with the exception of the co-imaging of MinD and MinE.

The double-labeled *minDE* strain used in this study, FW1919 (W3110 [*ΔminDE :: exobrs-sfGFP-minD minE-mKate2 :: frt*]), was constructed using the λ RED recombination method (Datsenko & Wanner, 2000) after we had observed that plasmidborne MinDE fusions are prone to overexpression in long-term experiments, and that imaging of CFP rather easily leads to photobleaching and

photodamage to the cells. To obtain this strain, strain FW1554 (W3110 [*ΔminDE :: exobrs-sfGFP-minD minE :: frt*]) (Wu *et al*, 2015a) was transformed with pKD46, and made electro-competent. A linear fragment containing the chloramphenicol gene amplified from pKD3 was transformed into the resulting strain to replace the *frt* scar, thus yielding strain FW1626 (W3110 [*ΔminDE :: exobrs-sfGFP-minD minE :: cat*]). FW1626 was then transformed with pKD46, made competent, and transformed with a linear fragment containing a *mKate2::aph frt* sequence amplified from plasmid pFWB019 to produce strain FW1639 (W3110 [*ΔminDE :: exobrs-sfGFP-minD minE-mKate2 :: aph frt*]). FW1639 was then cured of kanamycin resistance using a pCP20 plasmid as described previously (Datsenko & Wanner, 2000) to yield the final strain FW1919. This strain grows in rod shape in both M9 minimal medium and LB-rich medium, and produces no minicells, indicating that MinE-mKate2 is fully functional. However, both its fluorescence intensity and photostability in the cells are much lower than those of sfGFP-MinD, and thus less suitable for long-term imaging than the latter.

### Growth conditions

The M9-rich medium used previously (Wu *et al*, 2015b) and in the majority of the experiments in this study (unless specified) contained M9 salts, 0.4% glucose, and 0.25% protein hydrolysate amidase. The M9-poor medium contained M9 salts, 0.4% glucose, and 0.01% leucine. At 30°C, the doubling time of BN1590 cells during exponential growth was $104 \pm 9$ min in M9-poor liquid medium, and $69 \pm 3$ min in M9-rich liquid medium.

For cell shaping, cells were first inoculated into M9 liquid medium supplemented with 4 μg/ml A22 and incubated at 30°C for 3.5 h (rich medium) or 6 h (poor medium). The agarose pad used to seal the microchambers contained M9 medium supplemented with 4 μg/ml A22 and 25 μg/ml cephalexin as described previously. All cell-shaping experiments were carried out at 26°C.

### Cell shaping

The cell-sculpting method was used as described previously (Wu & Dekker, 2015; Wu *et al*, 2015b), with the following modifications. Prior to inoculation of the cells, the cover glass with the PDMS structures was treated with $O_2$ plasma for 10 sec to make the surface hydrophilic, which facilitates wetting of the surface and allows for more homogeneous inoculation of the cells into the microchambers. After the cells had settled into the microchambers, these were sealed with a small piece of agarose pad, as described previously (Wu *et al*, 2015b). We then poured 1 ml of warm agarose onto the existing agarose, which prevented the agarose from drying out during the long time course of the imaging. These two modifications in the cell-sculpting process increased the throughput of the shaping method, as well as minimizing the movement of the cells in the chambers due to drag of the drying agarose.

### Fluorescence microscopy

Fluorescence imaging was carried out with the same set-up as previously described (Wu *et al*, 2015b), but the following

modifications were introduced to facilitate long-term tracking. We used an upgraded perfect focus system (PFS3) on the Nikon Ti microscope, which has a larger *z*-range than the PFS2 system. While PFS3 was optimized for detecting the glass–water interface, we find that it can be used to locate the interface between glass and PDMS, which was then used to correct for the drift in *z* over time and keep the cells in focus. The PDMS layer with a thickness of 5–10 μm is within the sampling range for the PFS3, such that we can define the position of the cell with reference to the glass–PDMS interface. To track sfGFP-MinD during the whole course of cell growth, we used a time interval of 2 min. To monitor in detail the symmetry-breaking process that permits sfGFP-MinD patterns to emerge from homogeneity, we took fluorescence images sfGFP-MinD at intervals of 5–20 s, and only imaged the cytosol before and after this acquisition period. To examine the stability of the transverse oscillations, we used a 20-s time interval. To sample the effect of cell width on the final oscillation patterns in cells, we imaged every 5 min to obtain a larger dataset per experiment. Despite the fact that sfGFP is relatively resistant to photobleaching, it is critical to use low-intensity light for excitation in order to avoid photodamage to the cells, which reduces oscillation frequencies and eventually causes cell lysis.

### Image analysis

The cytosolic fluorescence images of the cells were processed in *Matlab* as described previously for boundary determination (Wu *et al*, 2015b). The binary image was used to define the lengths of the Feret diameters along the full 360° angular coordinates. From these data, the maximum and minimum Feret diameters were determined. The center of the MinD cluster was determined as described previously using a *Matlab* script (Wu *et al*, 2015b), and its angle was determined from its location relative to the cell center. The Feret diameter along this angle was used to compare the oscillation distance with the Feret diameters. Note that we use the Feret diameter along the oscillation angle as a measure of how well oscillations align with long or short axes, but this does imply that it represents a fair estimate of the distance traversed by each MinD protein. All the angle values extracted above are folded to between 0° and 90° due to the multifold symmetry of rectangles. Note that this MinD tracking method is restricted to the analysis of two-node oscillations and is not suitable for striped oscillations. The analyses of the final patterns in cells with various widths were carried out manually. The Matlab scripts used in this study are provided as Computer Code EV1.

### Analytical and numerical methods

All simulations were performed using the FEM method as implemented in the software *Comsol Multiphysics 4.4*. The linear stability analysis was performed with Wolfram *Mathematica 10* in elliptical geometry as introduced in Halatek and Frey (2012). We define the non-degeneracy of even and odd modes as:

$$d = \sqrt{\sum_{i=1}^{3} \left( Re(\sigma_i^e) - Re(\sigma_i^o) \right)^2}$$

where $Re(\sigma_i^e)$ and $Re(\sigma_i^o)$ denote the growth rate of the *i*-th even and odd mode, respectively.

The model is based on bulk–boundary coupling through a reactive boundary condition as introduced in Halatek and Frey (2012). For the cytosol, model equations read:

$$\partial_t u_{DD} = D_D \nabla^2 u_{DD} - \lambda u_{DD}$$
$$\partial_t u_{DT} = D_D \nabla^2 u_{DT} + \lambda u_{DD}$$
$$\partial_t u_E = D_E \nabla^2 u_E$$

Here, $u_{DD}$ denotes the density of cytosolic MinD-ADP, $u_{DT}$ cytosolic MinD-ATP, and $u_E$ cytosolic MinE; $\nabla$ the Nabla/Del operator in the cytosol (coordinate-free); $D_D$ the diffusion coefficient for cytosolic MinD, $D_E$ the diffusion coefficient for cytosolic MinE, and $\lambda$ the cytosolic nucleotide exchange rate.

At the membrane, we have

$$\partial_t u_d = D_m \nabla_m^2 u_d + (k_D + k_{dD} u_d) u_{DT} - k_{dE} u_d u_E$$
$$\partial_t u_{de} = D_m \nabla^2 u_{de} + k_{dE} u_d u_E - k_{de} u_{de}$$

Here, $u_d$ denotes the density of membrane-bound MinD, and $u_{de}$ membrane-bound MinDE complexes; $\nabla_m$ the Nabla/Del operator on the membrane (coordinate-free); $D_m$ the diffusion coefficient for the membrane, $k_D$ the MinD attachment rate constant, $k_{de}$ the MinDE detachment rate, $k_{dD}$ the MinD recruitment rate constant, $k_{dE}$ the MinE recruitment rate constant. Membrane and cytosolic dynamics are coupled by a system of reactive boundary conditions:

$$D_D \nabla_n u_{DD} = k_{de} u_{de}$$
$$D_D \nabla_n u_{DT} = -(k_D + k_{dD} u_d) u_{DT}$$
$$D_E \nabla_n u_E = -k_{dE} u_d u_E + k_{de} u_{de}$$

Here, $\nabla_n$ denotes the (outer) normal derivative at the boundary of the cytosol (membrane). Unless noted otherwise, all system parameters are taken from Halatek and Frey (2012), cf. listing in the Appendix.

**Expanded View** for this article is available online.

### Acknowledgements

F.W. and C.D. thank Rutger Hermsen, Jacob Kerssemakers, Felix Hol, and Juan Keymer for valuable discussions in the early stage of the research, and E. van Rijn for technical support. F.W. and C.D. are supported by the ERC Advanced Grant SynDiv (No. 669598), the Netherlands Organization of Scientific Research (NWO/OCW) as part of the Frontiers of Nanoscience Program, and the NanoNextNL Program 3B Nanomedicine. J.H. and E.F. are supported by the German Excellence Initiative via the NanoSystems Initiative Munich, and by the Deutsche Forschungsgemeinschaft (DFG) via Project B02 within SFB 1032 (Nanoagents for Spatio-Temporal Control of Molecular and Cellular Reactions).

### Author contributions

FW, JH, EF, and CD designed the work and wrote the paper. FW and EK carried out the experiments and analyzed the experimental data. JH performed the analytical and computational analysis of the model. MR implemented the automated numerical parameter sweeps. FW wrote the scripts for the analysis of experimental data.

### Conflict of interest

The authors declare that they have no conflict of interest.

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
