## [Review Process File · Molecular Systems Biology]

Multistability and dynamic transitions of intracellular Min protein patterns

Fabai Wu, Jacob Halatek, Matthias Reiter, Enzo Kingma, Erwin Frey and Cees Dekker

Corresponding authors: Cees Dekker, Delft University of Technology and Erwin Frey, Ludwig-Maximilians-Universität München

Review timeline:

Submission date:	02 December 2015
Editorial Decision:	13 January 2016
Revision received:	11 April 2016
Editorial Decision:	09 May 2016
Revision received:	13 May 2016
Accepted:	14 May 2016

Editor: Maria Polychronidou

Transaction Report:

1st Editorial Decision

13 January 2016

Thank you again for submitting your work to Molecular Systems Biology. We have now heard back from the three referees who agreed to evaluate your manuscript. Overall, the referees acknowledge that the presented findings seem potentially interesting. However, they raise a series of concerns, which should be carefully addressed in a revision of the manuscript.

Since the reviewers' recommendations are rather clear there is no need to repeat all the issues listed below. Some of the more fundamental issues raised by reviewers #2 and #3 refer to the need to demonstrate more convincingly the physiological and broader biological relevance of the main conclusions. These issues need to be satisfactorily addressed in a revision of the work. Moreover, along the lines of the comments of Reviewer #1, we would ask you to carefully edit the text in order to make sure that the main findings are easily accessible to a broad audience.

If you feel you can satisfactorily deal with these points and those listed by the referees, you may wish to submit a revised version of your manuscript. Please attach a covering letter giving details of the way in which you have handled each of the points raised by the referees. A revised manuscript will be once again subject to review and you probably understand that we can give you no guarantee at this stage that the eventual outcome will be favorable.

REFEREE COMMENTS

Reviewer #1:

Overall I am positive about the paper - it shows new and interesting results, and it puts these findings into a nice theoretical framework. It is well within the scope of the journal, and potentially interesting to a broad audience.

However, the main problem of the current manuscript is that it is hard to follow, for a number of different reasons:

- clarity of goals and discoveries
- overall structure
- phrases used to describe things

I urge the authors to re-evaluate the whole paper - possibly ask non-specialist colleagues to check it.

Confusing terms/phrases:

As far as I can figure out, the phrases "oscillation mode" and "stripe pattern" are both meant to signify dynamic oscillatory situations. The word "pattern", if not qualified (dynamic-pattern, oscillating pattern?) would mean to a naïve reader (like myself) a static pattern, while in fact none of these patterns are static.

Furthermore, "stripe" is confusing and (I would say) misleading. Sentences like "[the pattern] evolves into ... asymmetric oscillations, before converging into a stripe pattern". Again, this clearly implies it went from an oscillating situation to something else, ie. a non-oscillating situation.

"Stripe" implies something about shape (ie. a domain that is longer in one direction than another), but as far as I can see the domain in the oscillating case is also a stripe - especially in the simulated case (top row of Figure 2C).

I recognize that these are not criticisms of the research, but if the paper cannot be communicated clearly to a non-specialist audience then it would be more appropriate for a specialist journal.

Overall structure:

Despite phrases above being too ambiguous or specialist, many parts of the results section come across too much like the introduction, or something from a textbook, for example the paragraphs starting: "The classical account for symmetry breaking in chemical systems addresses...", and "To take the boundary geometry into account, the set of corresponding Fourier modes has to be derived first."

An attempt should be made to delineate clearly what has, and what has not, been achieved before - what is "common knowledge" - and as much of this should be in the introduction as possible.

Lack of succinct clarity:

Sentences such as the following are common in the paper, and are too convoluted to be helpful to the reader:

"The large effect this intrinsic bias has on initial pattern selection in a uniform system emphasizes the need to extend the approach based on uniform initial configurations when faced with multistable biological systems confined in bounded geometries by taking spatial heterogeneities into account."

Here's another example:

"The existence of a transverse mode is reminiscent of previous model predictions that additional destabilization of odd modes in elliptical geometry with low aspect ratio can explain circular and aberrant patterns in nearly spherical cells (Halatek & Frey, 2012)."

I would encourage simple clear sentences - especially for explaining the overall goals and discoveries. Would something like this not be possible in the introduction:

"Different dynamic modes have been seen (X, Y, Z), and cells are known to sometimes switch between these modes. Our goal was to understand which features influence the choice of these modes - both during the initial symmetry-breaking, and in the mode-switching. We studied the influence of intrinsic features such as reaction parameters, and imposed/external features such as geometry domain and perturbations. An important discovery was the high stability of a given mode to various perturbations - even in the face of significant geometry changes."

If this is correct, then it would have been nice to see a statement like this. If this is wrong, then I have misunderstood the paper!

Lack of clarity about the theoretical explanation:

This passage in section 2 does not seem true to me: "... in mathematical terms: if we decompose typical fluctuations around the uniform equilibrium in Fourier components, each component will set a characteristic length scale, but no particular component is preferred. Pattern formation is the selection and amplification of specific components from this spectrum. By means of a linear stability analysis, we can compute how fast each Fourier component will grow or decay with time when a perturbation is applied. Hence, symmetry breaking is a selection of the most unstable Fourier component from a broad, unbiased spectrum. For a planar geometry, this lack of bias translates directly to the linear stability analysis - there are no intrinsically preferred modes. However, in realistic system geometries, this holds no longer true in general."

However, many Turing systems have a preferred length scale independent of the domain geometry. In any case, the description appears to contradict this text from section 3: "Geometry sensing requires the existence of a characteristic length scale."

If this is a misunderstanding on my part, then it is another example of where the explanations should be written more clearly and carefully.

Reviewer #2:

In this work, Wu et al. studied the control of Min oscillations in *E. coli* focusing on the effect of geometry, dimensions, and initial spatial distributions of Min on the multistability of pattern formations by Min proteins in vivo. They find the MinD patterns are robust to various initial conditions and perturbations yet still dependent on spatial and kinetics parameters. Based on their previous theoretical work (Frey group), the authors made very good progress on exploring the geometry and kinetics parameter space to attempt to explain the origins and robustness of Min patterns. The thoroughness of the theoretical analysis is admirable, and the clear writing is a major strength of the manuscript and will be helpful to those in the future who want to survey the state-of-the-art in the field. Their experiments (Dekker lab) are clean-cut and informative by providing well-resolved spatial and temporal observations. Clearly their powerful microfluidics has tremendously helped the authors study the intracellular dynamics of proteins in live cells with altered geometry of confinement. Overall, this is nice collaborative work between biophysical modeling and experiment. We do, however, have comments that the authors should consider in a revised version before acceptance.

Major comments:

Although the authors undoubtedly consider the cell as a three-dimensional space ("For simulations of realistic 3D cellular geometries, ...", line 15, page 9), their system is essentially two-dimensional with the Z dimension being much smaller than X and Y. If they can consider a 3D geometry (e.g., XYZ being comparable) and sketch their expectations either qualitatively or quantitatively, that would be informative. Experiments are not needed considering the amount of work involved, although it would be interesting to see how the changes in the depth of shaped cells influence the pattern formations of Min proteins.

Experimentally, in general better physiological control is desirable. Since the nature of the work is biophysical, we would not insist investigating the physiology thoroughly in this work (e.g., effect of phytotoxicity, spatial confinement and cell-surface interactions etc on growth; whether the cells were growing in steady state etc). However, one relatively straightforward experiment would be the effect of growth conditions. If the authors can show that their experimental observations are robust to growth conditions by testing for at least one different growth medium, which we presume what the theory would predict, that would be very informative.

The work mainly focuses on MinD. However, when discussing symmetry breaking of homogeneous initial conditions, the MinE distributions might be spatially inhomogeneous. Since MinE fluorescent fusion proteins are available, two-color imaging would be an obvious experiment to perform and analyzed via modeling/theory.

To add a general comment extending point 3 above, while each of modeling and experiment components is well executed, the level of interplay between the two is comparison, not prediction. The prediction-level interactions would involve genetically and biochemically controlling KdD and KdE, making quantitative predictions, and testing them experimentally. This is obviously beyond the scope of the work and the authors' expertise, but something to consider in the future.

Minor comments:

In Figure 5 C, F, H, not seen error bars on the traces indicating the variation of the n cells. Also, these panels are painful to the eye.

In addition to one single cell exemplified in Fig. 5 B E and G, an overlaid pattern of all n cells would be better to be shown.

Scale bar is lacked in Fig. 1E.

More information about growth conditions need to be described (see also Major comment 2).

Reviewer #3:

This paper presents a systematic analysis of the spatiotemporal dynamics of the *E. coli* Min oscillation system, and of its dependence on the cell's geometry and on certain molecular properties. The bulk and relevance of the work is mostly theoretical, since the experimental results have mainly been published previously elsewhere by some of the authors. The paper offers an interesting analysis linking well-established methods from nonlinear physics with state-of-the-art measurements at the single-cell level, using a clever trick to control the cell's geometry (which again, has been published before). I personally like the approach very much as a tour de force in theoretical biology (with the benefit of experimental confirmation), although I have some doubts on the direct biological implications of the study:

-First, the experimental setup is rather artificial. The shape of *E. coli* cells (in particular their aspect ratio) is very well defined in natural conditions, whereas the shapes imposed by the authors (specially the ones offering spatiotemporal patterns different from simple pole-to-pole oscillations) are very far from real cells. On page 3, line 6, the authors mention diverse oscillation modes emerging during cell growth, but as far as I understand the oscillation mode does not usually change as the cell grows (see for instance Fischer-Friedrich et al, PNAS 2010, PMID 20308588).

-Second, the use of Min oscillations as a mechanism for cells to sense their geometry is rather restricted to *E. coli* cells. Not even other bacterial cells, let alone more complex cell types, use Min oscillations, thus the results obtain here seem somewhat limited.

-Third, the analytical and computational approach used in the paper forces the authors to consider homogeneous boundaries, with either relatively small (but unbiased) fluctuations, or at most a linear gradient in a specific membrane process (MinD attachment). However, as the authors themselves mention on page 7, line 36, many different factors can induce asymmetries and heterogeneities, in particular the sensitivity of lipid composition (and therefore MinD attachment) to curvature. This would lead to geometry-dependent factors that should severely affect the pattern selection process, and thereby might render the study presented here again very limited in scope.

In any case, as mentioned above I do find the study a valuable addition to the literature of biological pattern formation, and thus I would be happy to be shown that my concerns above are unfounded.

I would also like to raise the following minor issues:

1.- The authors use the term "Turing bifurcation" profusely throughout the text to refer to the emergence of their oscillating patterns, but that term is very well defined in the theory of pattern formation, where it is mainly used to refer to *stationary* patterns. In particular, this is the case of most of the papers cited by the authors when referring to Turing patterns. Thus I find their use of the term "Turing bifurcation" potentially confusing. The same comment applies for instance to the term "stripe pattern", used by the authors to refer to the "first harmonic" longitudinal mode. Usually stripe patterns denote stationary patterns, thus the notation might be confusing to the reader.

2.- As a technical curiosity, I was wondering how the authors avoid photobleaching effects when monitoring sfGFP-MinD at a time resolution of 20 seconds. I would expect other readers to have the same question.

3.- Starting on page 6, the authors mention the use of white noise as a spatially non-uniform perturbation of the homogeneous (or zero-order) spatial equilibrium state, when introducing the concept of Turing instability. I wonder whether they refer to noise white in time, white in space, or in both, and why is it necessary that the noise is white (in fact the noise is frequently correlated in time in cells, see for instance Rosenfeld et al, Science, 2005, PMID 15790856).

4.- The scheme in figure 3B is potentially misleading, specifically in the interval labeled "perturb and check stability". Do the authors imply that different gradient angles are applied sequentially in time? From the text I interpret that the angle is fixed in that part of the protocol.

5.- The authors refer several times throughout the paper (e.g. page 10, line 34) to "3D cell shapes", but as I understand all their simulations are done in 2D.

1st Revision - authors' response

11 April 2016

Reviewer #1:

Overall I am positive about the paper - it shows new and interesting results, and it puts these findings into a nice theoretical framework. It is well within the scope of the journal, and potentially interesting to a broad audience.

However, the main problem of the current manuscript is that it is hard to follow, for a number of different reasons:

- clarity of goals and discoveries*
- overall structure*
- phrases used to describe things*

I urge the authors to re-evaluate the whole paper - possibly ask non-specialist colleagues to check it.

We thank the review #1 for his positive comments on the manuscript and his general suggestion for improving the clarity of the text, which we have tried to accommodate in the revised manuscript. In particular, the text of the theory sections has been extensively edited throughout for clarity.

Confusing terms/phrases:

As far as I can figure out, the phrases "oscillation mode" and "stripe pattern" are both meant to signify dynamic oscillatory situations. The word "pattern", if not qualified (dynamic-pattern, oscillating pattern?) would mean to a naïve reader (like myself) a static pattern, while in fact none of these patterns are static.

Furthermore, "stripe" is confusing and (I would say) misleading. Sentences like "[the pattern] evolves into ... asymmetric oscillations, before converging into a stripe pattern". Again, this clearly implies it went from an oscillating situation to something else, ie. a non-oscillating situation. "Stripe" implies something about shape (ie. a domain that is longer in one direction than another), but as far as I can see the domain in the oscillating case is also a stripe - especially in the simulated case (top row of Figure 2C).

We inherit the term 'stripe' patterns from the previous literature of the Min system to describe the dynamic oscillations that have more than two nodes. As pointed out by the reviewer, it can introduce confusion to first-time readers. We have now, upon first use of the wording, expressed more clearly that striped modes are multi-node dynamic oscillations, see page 3 line 9. Furthermore, we have changed all occurrences of "stripe pattern" to "striped oscillations" throughout the text.

We have now also rephrased the sentence that the referee mentioned (page 5 lines 46-50).

I recognize that these are not criticisms of the research, but if the paper cannot be communicated clearly to a non-specialist audience then it would be more appropriate for a specialist journal.

Overall structure:

Despite phrases above being too ambiguous or specialist, many parts of the results section come across too much like the introduction, or something from a textbook, for example the paragraphs starting: "The classical account for symmetry breaking in chemical systems addresses...", and "To take the boundary geometry into account, the set of corresponding Fourier modes has to be derived first."

An attempt should be made to delineate clearly what has, and what has not, been achieved before - what is "common knowledge" - and as much of this should be in the introduction as possible.

Our original intention was to provide the required theoretical background (which is not common knowledge for a broad audience) at the start of the theory results sections. We are also concerned that putting all this material in the introduction section would cause interruptions in the line of the story.

We have discussed this issue with the editor and found a solution that we believe resolves the issue: we relegated the introductory part for the Turing instability to a dedicated text box (see page 7), which will be independently displayed in the article. We believe this will be helpful to readers unfamiliar to the topic.

Lack of succinct clarity:

Sentences such as the following are common in the paper, and are too convoluted to be helpful to the reader:

"The large effect this intrinsic bias has on initial pattern selection in a uniform system emphasizes the need to extend the approach based on uniform initial configurations when faced with multistable biological systems confined in bounded geometries by taking spatial heterogeneities into account."

This part has been rewritten for clarity; please see page 9 lines 38-40

Here's another example:

"The existence of a transverse mode is reminiscent of previous model predictions that additional destabilization of odd modes in elliptical geometry with low aspect ratio can explain circular and aberrant patterns in nearly spherical cells (Halatek & Frey, 2012)."

This part has been rewritten for clarity; please see page 10 lines 12-21.

I would encourage simple clear sentences - especially for explaining the overall goals and discoveries. Would something like this not be possible in the introduction:

"Different dynamic modes have been seen (X, Y, Z), and cells are known to sometimes switch between these modes. Our goal was to understand which features influence the choice of these modes - both during the initial symmetry-breaking, and in the mode-switching. We studied the influence of intrinsic features such as reaction parameters, and imposed/external features such as geometry domain and perturbations. An important discovery was the high stability of a given mode to various perturbations - even in the face of significant geometry changes."

If this is correct, then it would have been nice to see a statement like this. If this is wrong, then I have misunderstood the paper!

We thank the reviewer very much for this excellent suggestion. We have added a summary paragraph similar to the referee's suggestion to the introduction (page 4 lines 4 - 14). We have also adjusted the structure of the introduction to clarify the motivation of the work.

Lack of clarity about the theoretical explanation:

This passage in section 2 does not seem true to me: "... in mathematical terms: if we decompose typical fluctuations around the uniform equilibrium in Fourier components, each component will set a characteristic length scale, but no particular component is preferred. Pattern formation is the selection and amplification of specific components from this spectrum. By means of a linear stability analysis, we can compute how fast each Fourier component will grow or decay with time when a perturbation is applied. Hence, symmetry breaking is a selection of the most unstable Fourier component from a broad, unbiased spectrum. For a planar geometry, this lack of bias translates directly to the linear stability analysis - there are no intrinsically preferred modes. However, in realistic system geometries, this holds no longer true in general."

However, many Turing systems have a preferred length scale independent of the domain geometry. In any case, the description appears to contradict this text from section 3: "Geometry sensing requires the existence of a characteristic length scale."

If this is a misunderstanding on my part, then it is another example of where the explanations should be written more clearly and carefully.

This indeed seems to be misunderstanding, which needs to be clarified. We have now substantially rewritten this entire section that now comprises the text box on the Turing instability (page 7). This section is needed to set the stage for the analysis of two important and distinct questions: (i) What is the analogue of a homogeneous initial state for a mass-conserving reaction-diffusion system in confined geometry, and how does that initial state determined/influence the initial selection of a particular mode/pattern and hence length scale? (ii) What determines the length scale in the final spatio-temporal pattern?

Our main message in the text box section is that the initial phases of a "symmetry breaking" process in a nonlinear system is a *selection mechanism*. Consider an initial steady state of the corresponding well-mixed system that is weakly perturbed spatially, say by some spatially white noise. For the planar geometry considered in textbooks and review articles this is a spatially uniform state. The spectral decomposition of this state gives equal weight to all Fourier components and, therefore, sets no bias for a particular mode. Therefore, the dispersion relation obtained from a linear stability analysis of the reaction-diffusion equations for the system under consideration determine the fastest growing mode as the dominant mode in the initial pattern (Turing instability). This is textbook knowledge and well known. Our intention was to reiterate this background for a broad audience and contrast it with a generic situation encountered when one studies reaction-diffusion equations in confined intracellular space. There, as has been addressed recently in Thalmeier et al (PNAS 113 (3), 2016), the well-mixed state is generically *spatially non-uniform* for reaction-diffusion systems based on membrane-cytosol cycling and an NTPase activity. This non-uniformity in the initial state introduces a bias in mode selection which turns out to be a key aspect in our manuscript.

Now, while this box section deals with the initial stage of pattern formation due to the Turing instability, section 3 "Geometry sensing requires the existence of a characteristic length scale", addresses a genuinely different question, namely what determines the *final* pattern. Let us emphasize

that the initial selection of a length scale due to a Turing instability does not necessarily imply that this very same length scale also determines the final pattern! Mathematically speaking, the former is due to the interplay between the symmetry of the initial state and the form of the dispersion relation, while the latter emerges from the *nonlinear interaction* between different modes. Therefore, it is very well possible (and in many cases of intracellular dynamics quite generically so) that the final patterns are genuinely different and unrelated to the length scale set by the initial Turing instability. This is a major point we have made in our previous theoretical studies (Halatek & Frey, 2012), and which reiterates here in our combined experimental and theoretical analysis of the Min dynamics in different geometries. It is a major point which is easily overlooked, and we have now reemphasized it in the manuscript (page 7, lines 1-3) to make the point as clear as possible.

Let us give an example: A generic final pattern one finds in intracellular reaction-diffusion system is a pinned wavefront, which divides the entire system in two parts regardless of the system size. This pattern, which is reached for low MinD recruitment (no canalised transfer), does not convey any length scale since it remains invariant with respect to changes of system length. In other words, the pattern actually scales and adapts to cell length. Our simulations (c.f. Fig 3 C/D) indicate that this pattern always aligns with the long axis, hence it does not adapt to geometry in the sense that it prefers a certain range of length scales as the Turing type patterns do (obtained with high MinD recruitment, active canalised transfer).

We would like to note that a rigorous mathematical analysis of these important observations is possible, but far from trivial and beyond the scope of this paper, where we limit the exposition to the observations. A technical paper providing the broad theoretical background and justification is currently in preparation by the authors Halatek and Frey.

Reviewer #2:

In this work, Wu et al. studied the control of Min oscillations in E. coli focusing on the effect of geometry, dimensions, and initial spatial distributions of Min on the multistability of pattern formations by Min proteins in vivo. They find the MinD patterns are robust to various initial conditions and perturbations yet still dependent on spatial and kinetics parameters. Based on their previous theoretical work (Frey group), the authors made very good progress on exploring the geometry and kinetics parameter space to attempt to explain the origins and robustness of Min patterns. The thoroughness of the theoretical analysis is admirable, and the clear writing is a major strength of the manuscript and will be helpful to those in the future who want to survey the state-of-the-art in the field. Their experiments (Dekker lab) are clean-cut and informative by providing well-resolved spatial and temporal observations. Clearly their powerful microfluidics has tremendously helped the authors study the intracellular dynamics of proteins in live cells with altered geometry of confinement. Overall, this is nice collaborative work between biophysical modeling and experiment. We do, however, have comments that the authors should consider in a revised version before acceptance.

We thank reviewer #2 for the positive comments and we note that he/she particularly appreciated the clear writing as a major strength of the manuscript.

Major comments:

Although the authors undoubtedly consider the cell as a three-dimensional space ("For simulations of realistic 3D cellular geometries, ...", line 15, page 9), their system is essentially two-dimensional with the Z dimension being much smaller than X and Y. If they can consider a 3D geometry (e.g., XYZ being comparable) and sketch their expectations either qualitatively or quantitatively, that would be informative. Experiments are not needed considering the amount of work involved, although it would be interesting to see how the changes in the depth of shaped cells influence the pattern formations of Min proteins.

We thank the referee for this interesting point. To address it, we ran a sequence of simulations where the height of the cell was increased. For the same kinetic parameter configurations (canalized transfer) where we found multi-stability in cells with low height, we found that additional patterns in the z-direction become active if the height is moved in the same range as the minimal width that

renders traversal modes active ($\sim 3\text{-}4\mu\text{m}$). In a representative cell with cell dimensions of $5\times 4\times 3.1\mu\text{m}^3$, we verified side-to-side oscillations that could be aligned to all three axes. Since our computational framework was specifically designed to assess the competition between longitudinal and traversal modes, a straightforward extension of the analysis to include patterns in z-direction is not feasible in the given time. It would, however, be very interesting to address this question in a future research project (potentially also including experiments).

We have included this additional result now in the main text at page 13 lines 8-16. We conclude that increasing the height increases the diversity of distinct patterns even further.

Experimentally, in general better physiological control is desirable. Since the nature of the work is biophysical, we would not insist investigating the physiology thoroughly in this work (e.g., effect of phytotoxicity, spatial confinement and cell-surface interactions etc on growth; whether the cells were growing in steady state etc). However, one relatively straightforward experiment would be the effect of growth conditions. If the authors can show that their experimental observations are robust to growth conditions by testing for at least one different growth medium, which we presume what the theory would predict, that would be very informative.

Again, we find the suggestion very useful, and we have now run several experiments with M9 minimum medium (in contrast to M9 rich medium), in which the cells are growing slower but nevertheless are able to grow into a size of about $9\times 5\times 1\mu\text{m}^3$. Indeed, as expected, all the phenomena described in the paper were also observed in this new M9 minimum medium. The additional results are now shown in the insets of Fig. 6C and 6E, and are described in the text page 15 line 31-41.

The work mainly focuses on MinD. However, when discussing symmetry breaking of homogeneous initial conditions, the MinE distributions might be spatially inhomogeneous. Since MinE fluorescent fusion proteins are available, two-color imaging would be an obvious experiment to perform and analyzed via modeling/theory.

Imaging of MinDE has always been tricky due to the lower signal-to-noise ratio of the MinE signal. However, we have now constructed a strain co-expressing fluorescent fusions of MinD and MinE. The construction of the strain is described in the methods section and the result is shown in the form of Movie EV3, and described in page 6 line 13-20. As we did not find any surprising MinE distribution that dictates the initial MinD symmetry breaking, we opt not to describe the results in unnecessary detail.

To add a general comment extending point 3 above, while each of modeling and experiment components is well executed, the level of interplay between the two is comparison, not prediction. The prediction-level interactions would involve genetically and biochemically controlling KdD and KdE, making quantitative predictions, and testing them experimentally. This is obviously beyond the scope of the work and the authors' expertise, but something to consider in the future.

We thank the referee for all these valuable suggestions and hope that indeed future genetic/biochemical work will provide us with means to probe the effect of these parameters experimentally.

Minor comments:

In Figure 5 C, F, H, not seen error bars on the traces indicating the variation of the n cells. Also, these panels are painful to the eye.

In addition to one single cell exemplified in Fig. 5 B E and G, an overlaid pattern of all n cells would be better to be shown.

There seems to be a misunderstanding caused by lack of clarification from our side. The plots in Fig. C, F, and H are quantitative data extracted from single cells. The numbers merely indicate how many cells in the full set of data show such behavior. Because each cell grows differently and has transitions at different time point, plotting them together will be very messy. The same applies to the Fig. 5B, E, and G. We have now provided more examples in the form of Movie EV4, and we think that this should be sufficient for the readers to capture the similarities and differences between cells. In order to avoid confusion, we have now added clarifications in the caption text and we have

moved the number of cells to the caption.

Scale bar is lacked in Fig. 1E.

Fig. 1E shares the scale bar with Fig. 1B-D. We have now clarified this in the caption.

More information about growth conditions need to be described (see also Major comment 2).

We have now added detailed descriptions of the nutrient conditions in the methods section. We previously skipped the detailed description because, except for the new experiments with M9 minimum medium, we did use the same growth condition as *Wu et al 2015 Nature Nanotechnol.*, which we referred to.

Reviewer #3:

This paper presents a systematic analysis of the spatiotemporal dynamics of the E. coli Min oscillation system, and of its dependence on the cell's geometry and on certain molecular properties. The bulk and relevance of the work is mostly theoretical, since the experimental results have mainly been published previously elsewhere by some of the authors. The paper offers an interesting analysis linking well-established methods from nonlinear physics with state-of-the-art measurements at the single-cell level, using a clever trick to control the cell's geometry (which again, has been published before). I personally like the approach very much as a tour de force in theoretical biology (with the benefit of experimental confirmation), although I have some doubts on the direct biological implications of the study:

We thank the reviewer for the positive comments on the manuscript. We would however like to point out that all the experimental results reported here are new and unpublished, and go very significantly beyond the general experimental approach from our previous paper (*Wu et al 2015 Nature Nanotechnology*). In that paper, the limited imaging time (2 minutes for each single cell) allowed to identify various oscillation modes. But it was insufficient to address important questions concerning the long-time persistence and selection of patterns, nor could we trace temporal transitions between patterns. The improvement that we have implemented and described in the experimental setup and methods in the current study (see the Material and methods section) were instrumental for the success to overcome the previous limitations. With these technical improvements, we are now able to capture the symmetry breaking, the robustness of the oscillations, and the dynamic transitions. Indeed, these three points are central aspects of the current paper.

-First, the experimental setup is rather artificial. The shape of E. coli cells (in particular their aspect ratio) is very well defined in natural conditions, whereas the shapes imposed by the authors (specially the ones offering spatiotemporal patterns different from simple pole-to-pole oscillations) are very far from real cells.

This is correct – these shapes are indeed artificial. However, the interest of these anomalous shapes is that they allow to explore phase space in a manner that would not be possible in more traditional cell biology approaches. Using our shaped cells we are able to probe for general principles about reaction-diffusion patterns that was current unaccessible in other systems.

In the introduction, we stressed the current gap between the general pattern formation theory (including computational approaches) and the physiological relevance. Scanning through literature, there is barely another Turing system in biology as well studied as the Min system. The previous extensive understanding of the Min system provide us with the possibility to study highly complex pattern formation processes in quantitative detail. We believe that the gained insights are very relevant to biological systems: e.g. the symmetry breaking out of equilibrium, the effect of spatial heterogeneity, and the effect of cell geometry in the context of growth (see paragraph 2). Moreover, the framework that emerges from our study has general applicability to other intracellular pattern forming systems based on reaction-diffusion.

On page 3, line 6, the authors mention diverse oscillation modes emerging during cell growth, but as far as I understand the oscillation mode does not usually change as the cell grows (see for

instance Fischer-Friedrich et al, PNAS 2010, PMID 20308588).

We have slightly rephrased the mentioned line to make it more accurate. See page 3 lines 43-45.

Concerning the particular reference indicated, we would like to note that the long-time kymographs in the Fischer-Friedrich paper were obtained by massive overexpression. In fact, the main finding of the paper (stochastic switching) has been identified as an overexpression artifact (Sliusarenko et al, Mol Microbiol 2011, PMID 21414037). A comparison to our results is therefore not possible.

-Second, the use of Min oscillations as a mechanism for cells to sense their geometry is rather restricted to E. coli cells. Not even other bacterial cells, let alone more complex cell types, use Min oscillations, thus the results obtain here seem somewhat limited.

The Min system, especially MinD, is actually conserved in all three kingdoms of life. Carried over from their ancestral cyanobacterial endosymbiont, plants use MinDE to control the division of chloroplasts, and recently it was found that lots of eukaryotes use MinDE to control the division of mitochondria. Recently reported bacteria that divide along their longest axis have a MinCDE operon, despite the fact that they divide completely differently compared to *E. coli*. We have now added a first line in paragraph 3 of the Introduction section and listed a range of articles that refer to the above examples and more.

Just as the mechanism of most pattern formation system in nature is unknown, the behavior and mechanism of MinCDE in other organisms are mostly simply unknown. It may be that many bacteria that grow into different shapes and divide in different modes are all based on homologs of the *E. coli* MinCDE system, which provides yet another motivation to understand the *E. coli* Min system in different cell shapes.

Regarding the general implications of our studies on the Min system on non-oscillating positioning systems, we would like to emphasize that the core processes responsible for the localization of Min is universal: nonlinear kinetics and NTPase cycling between membrane and cytosol. While many systems do not show oscillatory behavior, their proteins constantly undergo dynamic recycling between cytosol and membrane, and thus they are by no means static. To emphasize the generality of the two above core processes, we added page 3 line 38-42.

-Third, the analytical and computational approach used in the paper forces the authors to consider homogeneous boundaries, with either relatively small (but unbiased) fluctuations, or at most a linear gradient in a specific membrane process (MinD attachment). However, as the authors themselves mention on page 7, line 36, many different factors can induce asymmetries and heterogeneities, in particular the sensitivity of lipid composition (and therefore MinD attachment) to curvature. This would lead to geometry-dependent factors that should severely affect the pattern selection process, and thereby might render the study presented here again very limited in scope.

Of course, strong geometric cues are always a candidate for intracellular patterns. However, in case of the Min system (and many other reaction-diffusion systems) pattern formation is self-organised and there is no experimental indication that pattern selection is guided by specific intracellular cues that bypass the effect of geometry that we are discussing. We have chosen the MinD attachment rate as control parameter of the spatial heterogeneity, because there is evidence that variations in the lipid composition are mainly reflected in the MinD attachment process. If clusters of anionic phospholipids that prefer high curvature would determine the pattern, higher-order longitudinal modes would be suppressed, and wave nodes would generally be expected to co-localise with the parts of higher curvature. However, this is not observed experimentally. In Halatek&Frey (Cell Rep 2012), we address this question and show that stripe formation is robust against heterogeneities in the MinD attachment and proteins can even accumulate at positions where MinD attachment is turned off entirely. This is possible because a low basal density of membrane-bound MinD can be achieved by lateral membrane diffusion, and accumulation is dominated by the MinD recruitment pathway which is independent of the MinD membrane affinity. The decision to use MinD attachment as control variable is entirely based on properties of the Min system. We do show that heterogeneities could have an effect on the selection of patterns but barely on their stability. This aspect has previously not been taken into account in the literature on intracellular pattern formation.

In any case, as mentioned above I do find the study a valuable addition to the literature of biological pattern formation, and thus I would be happy to be shown that my concerns above are unfounded.

We appreciate this opinion of the referee and we do hope that the above answers have been clarifying.

I would also like to raise the following minor issues:

*1.- The authors use the term "Turing bifurcation" profusely throughout the text to refer to the emergence of their oscillating patterns, but that term is very well defined in the theory of pattern formation, where it is mainly used to refer to *stationary* patterns. In particular, this is the case of most of the papers cited by the authors when referring to Turing patterns. Thus I find their use of the term "Turing bifurcation" potentially confusing. The same comment applies for instance to the term "stripe pattern", used by the authors to refer to the "first harmonic" longitudinal mode. Usually stripe patterns denote stationary patterns, thus the notation might be confusing to the reader.*

The possibility of oscillatory patterns was discussed by Alan Turing in his seminal paper. Hence, we found it appropriate to refer to the oscillatory instabilities as Turing instabilities as well. In particular, the underlying principle (destabilization by diffusive coupling) is the same as in the non-oscillatory case. Of course, we are aware that Turing patterns and Turing instabilities typically refer to the non-oscillatory case in the literature. To avoid any confusion in this regard, we now generally refer to "oscillatory Turing instabilities". Furthermore, we changed all occurrences of "stripe pattern" to "striped oscillations" – see our response to referee 1.

2.- As a technical curiosity, I was wondering how the authors avoid photobleaching effects when monitoring sfGFP-MinD at a time resolution of 20 seconds. I would expect other readers to have the same question.

This can be achieved owing to the superior brightness and photostability of sfGFP-MinD. We have now added lines 1-5 at page 5, as well as a few lines in the Material and methods section, to emphasize this point. There is trade off between signal and frequency of imaging. For long-term tracking with 20-second interval, we used very weak exposure intensities, which is why movie EV1 and Fig. 1F is very grainy (i.e. has a low signal-to-noise ratio). With 2-minute intervals, we were able to use higher light intensity to obtain very good signals. We also note now in the Material and methods section that photodamage to the cells is an even more important issue to avoid than photobleaching of the fluorescent proteins.

3. Starting on page 6, the authors mention the use of white noise as a spatially non-uniform perturbation of the homogeneous (or zero-order) spatial equilibrium state, when introducing the concept of Turing instability. I wonder whether they refer to noise white in time, white in space, or in both, and why is it necessary that the noise is white (in fact the noise is frequently correlated in time in cells, see for instance Rosenfeld et al, Science, 2005, PMID 15790856).

We refer to the noise in the context of the perturbation of the initial condition in the simulations. In this regard, it is a spatial noise and it is only needed to stimulate the growth of unstable modes as now described in the new text box (page 7).

4.- The scheme in figure 3B is potentially misleading, specifically in the interval labeled "perturb and check stability". Do the authors imply that different gradient angles are applied sequentially in time? From the text I interpret that the angle is fixed in that part of the protocol.

This was indeed misleading. The angles of the perturbations do not change in time but differ between simulations. We have now corrected the figure, and in the updated version we only show one representative perturbation template to avoid this confusion.

5.- The authors refer several times throughout the paper (e.g. page 10, line 34) to "3D cell shapes", but as I understand all their simulations are done in 2D.

This seems to be a misunderstanding. All simulations were performed in 3D geometries that resemble the cell shapes in the experiments. They are only presented as 2D projections for the

convenience of visualization. Only the linear stability analysis was performed in 2D elliptical geometry, since 3D geometries are (yet) analytically inaccessible.

2nd Editorial Decision

09 May 2016

Thank you again for submitting your work to Molecular Systems Biology. We have now heard back from two of the three referees who agreed to evaluate your manuscript. As you will see below, reviewers #2 and #3 think that the revised manuscript is suitable for publication. Since the comments of reviewer #1 mostly referred to the need to revise the text, which we and the other two reviewers think has now been improved, we do not see a reason to further delay our decision by waiting for the comments of Reviewer #1.

Before we can formally accept the manuscript for publication, we would like to ask you to address ~~the following~~ a few editorial issues listed below.÷

~~-In p. 22 you refer to the Appendix, however no Appendix PDF has been provided with the revised manuscript. Could you please provide the file?~~

~~-Please provide the Matlab codes used for data analysis as "Computer Code EV1" in a .zip file.~~

REFeree COMMENTS

Reviewer #2:

The revised manuscript fully addresses our concerns and we recommend its publication.

Reviewer #3:

In my opinion the authors have answered the referees' comments adequately.

Corresponding Author Name: Cees Dekker
Manuscript Number: MSB-15-6724R